# A dormant overmassive black hole in the early Universe

Ignas Juodžbalis[1,2 ✉], Roberto Maiolino[1,2,3,22], William M. Baker[1,2,22], Sandro Tacchella[1,2,22], Jan Scholtz[1,2,22], Francesco D'Eugenio[1,2,22], Joris Witstok[1,2,22], Raffaella Schneider[3,4,5,6,22], Alessandro Trinca[3,4,5,22], Rosa Valiante[4,5,22], Christa DeCoursey[7,22], Mirko Curti[8], Stefano Carniani[9], Jacopo Chevallard[10], Anna de Graaff[11], Santiago Arribas[12], Jake S. Bennett[13], Martin A. Bourne[1,14,15], Andrew J. Bunker[10], Stéphane Charlot[16], Brian Jiang[1], Sophie Koudmani[1,14,17,18], Michele Perna[12], Brant Robertson[19], Debora Sijacki[1,14], Hannah Übler[1,2], Christina C. Williams[20] & Chris Willott[21]

Recent observations have found a large number of supermassive black holes already in place in the first few hundred million years after the Big Bang, many of which seem to be overmassive relative to their host galaxy stellar mass when compared with local relation[1–9]. Several different models have been proposed to explain these findings, ranging from heavy seeds to light seeds experiencing bursts of high accretion rate[10–16]. Yet, current datasets are unable to differentiate between these various scenarios. Here we report the detection, from the JADES survey, of broad Hα emission in a galaxy at $z = 6.68$, which traces a black hole with a mass of about $4 \times 10^8 M_\odot$ and accreting at a rate of only 0.02 times the Eddington limit. The black hole to host galaxy stellar mass ratio is about 0.4—that is, about 1,000 times above the local relation—whereas the system is closer to the local relations in terms of dynamical mass and velocity dispersion of the host galaxy. This object is most likely an indication of a much larger population of dormant black holes around the epoch of reionization. Its properties are consistent with scenarios in which short bursts of super-Eddington accretion have resulted in black hole overgrowth and massive gas expulsion from the accretion disk; in between bursts, black holes spend most of their life in a dormant state.

The galaxy JADES GN+189.09144+62.22811 1001830 (hereafter GN-1001830), located in the GOODS-N field, was observed with JWST both with Near Infrared Camera (NIRCam) and with the Near Infrared Spectrograph (NIRSpec) multi-object mode, both with the low-resolution prism and medium-resolution gratings as part of the JADES (JWST Advanced Extragalactic Survey), PID:1181. The NIRSpec spectra reveal multiple emission nebular lines (Extended Data Fig. 1), which show that the galaxy is at $z = 6.677 \pm 0.004$.

The Hα line is in the gap of the medium-resolution grating spectrum and observed only in the prism spectrum (Fig. 1). However, the resolution at this wavelength is sufficient to reveal a clear broad component of this line. The broad component is fairly symmetric and not seen in [OIII] (Extended Data Fig. 3). This suggests that the broad Hα line is not associated with outflows, leaving as the most plausible interpretation the broad-line region (BLR) of an accreting black hole, that is, an active galactic nucleus (AGN).

The broad component has a width of $5,700^{+1,700}_{-1,100}$ km s$^{-1}$ and a flux of $27.3^{+4.1}_{-4.0} \times 10^{-19}$ erg s$^{-1}$ cm$^{-2}$. Assuming the local virial relations[17,18] and taking into account the effect of dust obscuration, we estimate a black hole mass of $\log(M_{BH}/M_\odot) = 8.61^{+0.38}_{-0.37}$ (Methods).

Coupled with the bolometric luminosity, estimated from the broad component of Hα, but also consistently from the photometric fit of the nuclear component (Methods), we infer that the galaxy is accreting at 2.4% of its Eddington limit, that is, $\lambda_{Edd} \equiv L_{bol}/L_{Edd} = 0.024^{+0.011}_{-0.008}$, with an intrinsic scatter of 0.5 dex (see Methods for details).

The fact that the AGN is so underluminous allows constraining the properties of the host galaxy much better than in luminous quasars. We used the ForcePho (Johnson, B., manuscript in preparation) tool to decompose the contribution of the nuclear region, hosting the unresolved AGN, from the host galaxy in NIRCam images (Methods). The morphology can be well fitted with a nuclear unresolved source and a

[1]Kavli Institute for Cosmology, University of Cambridge, Cambridge, UK. [2]Cavendish Laboratory - Astrophysics Group, University of Cambridge, Cambridge, UK. [3]Dipartimento di Fisica, 'Sapienza' Università di Roma, Roma, Italy. [4]Osservatorio Astronomico di Roma, INAF, Monte Porzio Catone, Italy. [5]INFN, Sezione Roma1, 'Sapienza' Università di Roma, Roma, Italy. [6]Sapienza School for Advanced Studies, Roma, Italy. [7]Steward Observatory, University of Arizona, Tucson, AZ, USA. [8]European Southern Observatory, Garching, Germany. [9]Scuola Normale Superiore, Pisa, Italy. [10]Department of Physics, University of Oxford, Oxford, UK. [11]Max-Planck-Institut für Astronomie, Heidelberg, Germany. [12]Centro de Astrobiología (CAB), CSIC-INTA, Madrid, Spain. [13]Center for Astrophysics, Harvard University, Cambridge, MA, USA. [14]Institute of Astronomy, University of Cambridge, Cambridge, UK. [15]Centre for Astrophysics Research, Department of Physics, Astronomy and Mathematics, University of Hertfordshire, Hatfield, UK. [16]Institut d'Astrophysique de Paris, Sorbonne Université, CNRS, Paris, France. [17]St Catharine's College, University of Cambridge, Cambridge, UK. [18]Center for Computational Astrophysics, Flatiron Institute, New York NY, USA. [19]Department of Astronomy and Astrophysics, University of California, Santa Cruz, CA, USA. [20]NSF's National Optical-Infrared Astronomy Research Laboratory, Tucson, AZ, USA. [21]NRC Herzberg, Victoria British Columbia, Canada. [22]These authors contributed equally: Roberto Maiolino, William M. Baker, Sandro Tacchella, Jan Scholtz, Francesco D'Eugenio, Joris Witstok, Raffaella Schneider, Alessandro Trinca, Rosa Valiante, Christa DeCoursey. ✉e-mail: ij284@cam.ac.uk

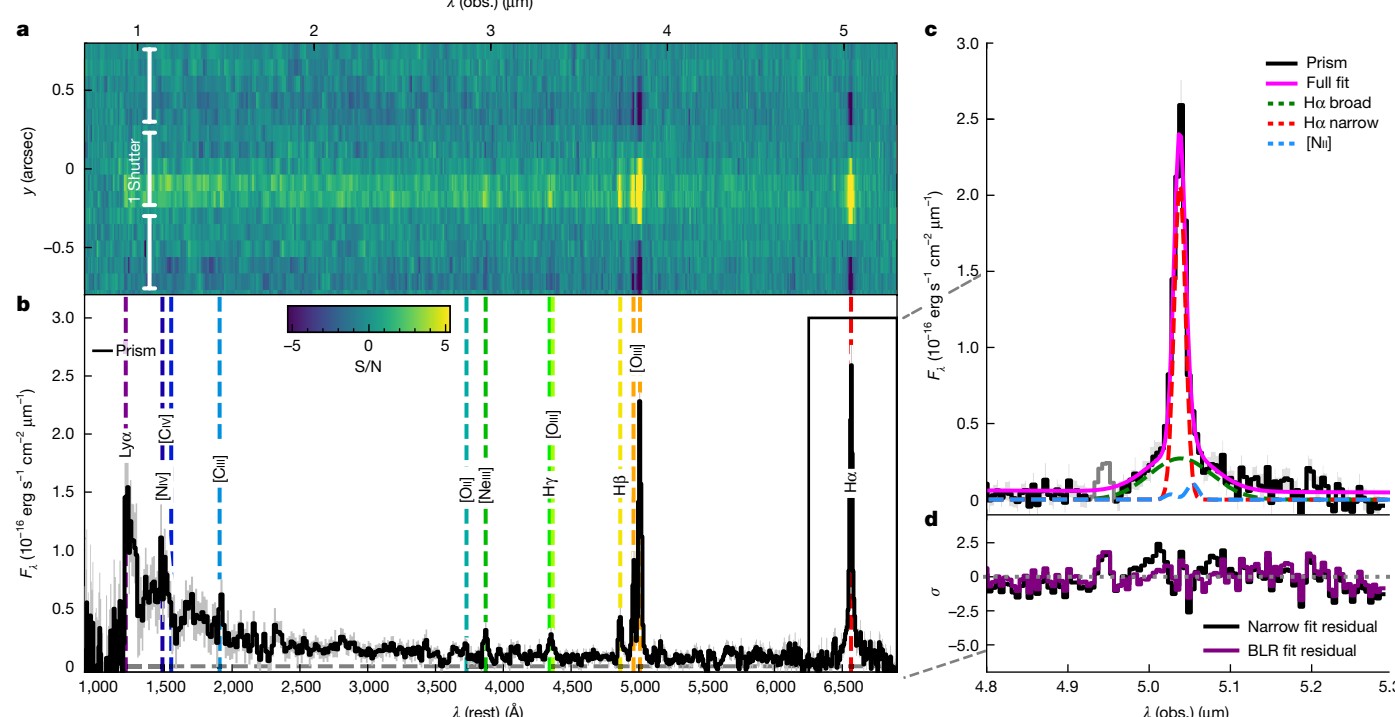

**Fig. 1 | Prism spectrum and Hα line of GN-1001830. a,** The two-dimensional prism spectrum. **b,** The one-dimensional prism (black line in the bottom panel) with marked emission lines. **c,** The spectrum around Hα showing the presence of a broad component. The lines shown are the observed spectrum (black solid line, with grey shading indicating $1\sigma$ uncertainties) along with the best-fit line to the narrow (red dashed) and broad (green dashed) components. The [NII] doublet is shown in blue; it is only marginally detected at $2\sigma$. The magenta solid line shows the total fit. The grey line portion at around 4.95 µm of the spectrum shows the region that was masked because of a possible artefact or Hα emission from a lower redshift interloper. **d,** The fit residuals for a simple narrow Hα and [NII] fit (black line) and the best fit, containing a broad component (purple line). The narrow-line-only fit does not account for the broad wings of the line, leaving substantial systematic residuals.

compact (with half-light radius $R_e \approx 140$ pc) host galaxy with a disc-like profile (Sérsic index $n \approx 1$).

The eight-band photometry of the host galaxy was then fitted with the SED fitting codes BAGPIPES[19] and Prospector[20] (Methods). These fits are consistent and averaged together, yielding a stellar mass $\log(M_*/M_\odot) = 8.92^{+0.30}_{-0.31}$ and instantaneous star formation rate SFR = $1.38^{+0.92}_{-0.45}$ $M_\odot$ yr$^{-1}$ (within the past 10 Myr), which places our object a factor of 3 below the star-forming main sequence at its redshift.

Figure 2 shows the location of GN-1001830 (large magenta circle) on the $L/L_{Edd}$ versus $M_{BH}$ diagram (Fig. 2a,c) and on the $M_{BH}$ versus $M_*$ diagram (Fig. 2b,d). In Fig. 2a,c, our source is compared with other AGN discovered by previous JWST studies at similar redshifts ($4 < z < 11$, blue symbols)[1–8,21], along with bright $z > 5$ quasi-stellar objects (QSOs) observed with JWST (orange and yellow symbols)[22–26]. Figure 2a shows that our object is among the most massive black holes found by JWST, with a mass similar to that of luminous high redshift quasars, but it accretes at a rate lower by about two orders of the observed negative correlation of magnitude. Therefore, our object is the dormant counterpart of luminous, high redshift quasars.

Moreover, Fig. 2b indicates that our object is one of the most overmassive black holes found by JWST, that is, the black hole mass approaches 50% of the stellar mass of the host—about 1,000 times above the local relation between the black hole and host galaxy stellar mass.

The JWST finding of several overmassive black holes at high redshift[1,2,4,9,27] has been interpreted by previous works[28,29] as the result of a large scatter of the black hole–stellar mass relation combined with selection effects, that is, more massive black holes tend to be preferentially selected, as they can reach higher luminosities. Our discovery of these overmassive black holes associated with a low luminosity AGN, because of its low Eddington ratio, is incompatible with the selection

effect scenarios as our data are deep enough to be less sensitive to selection effects (Fig. 3). This is discussed more extensively in the Methods.

Some previous studies have found that early black holes are overmassive only relative to the stellar mass, but when compared with the velocity dispersion and dynamical mass of the host galaxy, they are more aligned with the local relation[2,30]. As detailed in the Methods, based on the profile of the [OIII] doublet, we find that this is the case also for GN-1001830. In contrast to its strong offset on the $M_B$−$M_*$, this galaxy is closer to the local $M_{BH}$−$\sigma$ and $M_{BH}$−$M_{dyn}$ relations. This indicates that the baryonic mass of the host galaxy is already in place but that star formation lags, possibly because of feedback generated by black hole accretion.

The presence of overmassive black holes in the early Universe has been explained by a variety of models and cosmological simulations. These predict that black holes are born either from relatively massive seeds (often called heavy seeds, such as direct collapse black holes, originating from clouds of pristine gas) accreting below the Eddington rate or from short phases of super-Eddington accretion (possibly driven by galaxy mergers) either on light (stellar remnants) or on heavy seeds[10–16,31–34]. Figure 2c,d shows the comparison of GN-1001830 with the CAT semi-analytical models from refs. 10,11, which predict both scenarios, in a snapshot at $z = 7$. The Eddington-limited, heavy-seeds scenario (grey small symbols) fails to reproduce the properties of GN-1001830. To reproduce the large black hole masses observed at high redshift without exceeding the Eddington limit, this scenario requires black holes to be accreting close to the Eddington limit most of the time. Therefore, the high-mass black holes, predicted by this scenario, are not found at the highly sub-Eddington rates as observed in our object. Moreover, this scenario can reproduce either a very overmassive nature of black holes ($M_{BH}/M_* \approx 0.1$) only for low-mass galaxies ($M_* \sim 10^6$–$10^7 M_\odot$)

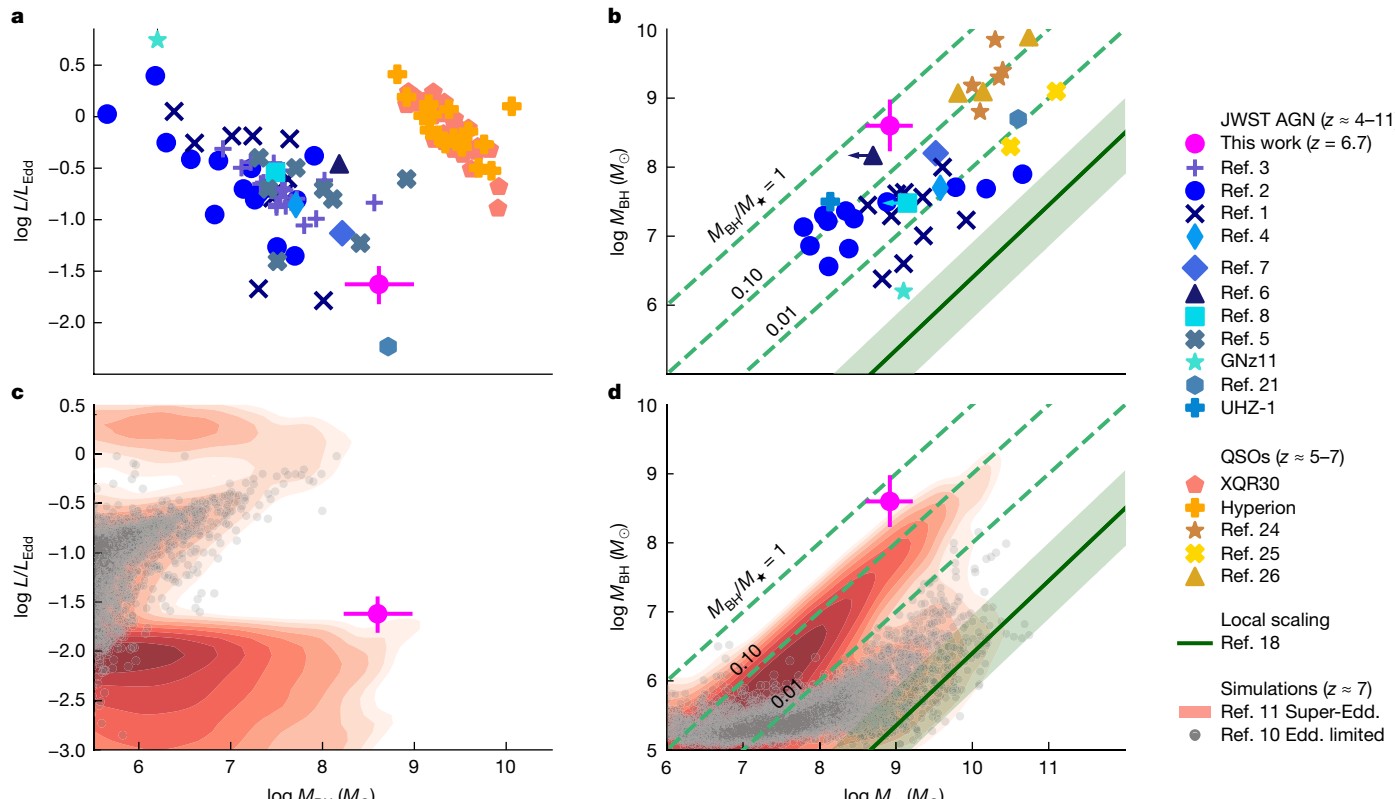

**Fig. 2 | Comparison of GN-1001830 with other high-z AGN and models in terms of accretion rate, black hole mass and stellar mass of the host galaxy.** **a,c**, Accretion rate relative to the Eddington limit, $\lambda_{Edd}$, versus black hole mass, $\log M_{BH}$. **b,d**, Black hole mass versus stellar mass of the host galaxy $\log M_*$. The green dashed lines indicate constant $M_{BH}/M_*$ ratios, whereas the solid green line represents the local relation from ref. 18; the shaded region shows the scatter. In all panels, GN-1001830 is indicated by a magenta circle with error bars. In **a** and **b**, comparison with other JWST-discovered AGN at high redshift is shown in blue[1–8,21] and with the QSO population at similar redshifts is shown in

orange and yellow[22–26]. The observed negative correlation between $\lambda_{Edd}$ and $M_{BH}$ is probably reflective of the dependence of Eddington luminosity on black hole mass and observational incompleteness and not a separate physical phenomenon. In **c** and **d**, comparison with the predictions (at $z \approx 7$) from the semi-analytical models from refs. 10,11 in the scenario of Eddington-limited accretion is shown as grey points and the scenario of light or heavy seeds that can experience super-Eddington accretion as red contours. Error bars indicate 1σ uncertainties.

or more moderately overmassive black holes ($M_{BH}/M_* \approx 0.01$) in more massive galaxies. Therefore, it is not able to reproduce the $M_{BH}/M_* = 0.43$ observed in GN-1001830 with $M_* = 2 \times 10^9 M_\odot$. By contrast, the models show that even starting from light seeds, allowing super-Eddington accretion bursts (red contours) can reproduce the observed properties of our object. It may sound counterintuitive that super-Eddington scenarios can better reproduce the relatively quiescent AGN in GN-1001830. The fact is that super-Eddington accretion phases allow the black hole to grow rapidly in short (1–4 Myr) bursts, whereas the resulting strong feedback makes the black hole lack gas to accrete significantly for long periods. Therefore, black holes can reach high masses while staying dormant for long periods, increasing the probability of seeing them in a low luminosity (dormant) state.

We note that the same result is also found when comparing with fully self-consistent cosmological simulations of galaxy formation, such as FABLE (Feedback Acting on Baryons in Large-scale Environments), as discussed in the Methods.

Apart from the super-Eddington scenario described above, models that implement radiatively inefficient accretion onto low-spin black holes[35] could also help explain our finding. However, a detailed treatment of this scenario is beyond the scope of our work.

It is tempting to speculate that our result favours light seed models. However, the same result would also hold if the models had started with heavy seeds. The key feature that allows the properties of GN-1001830 to be matched is the fact that accretion goes through super-Eddington phases, regardless of the seeding mechanism.

Finally, we argue that dormant, overmassive black holes in galaxies with low SFR, such as GN-1001830, are probably quite common in the early Universe. Finding one of them out of 35 spectroscopically targeted galaxies at $z > 6$ in the GOODS-N field, in a single tier of the JADES survey, is remarkable, as the JADES selection function at $z > 6$ disfavours the selection of high-z galaxies with low SFRs[36]. Moreover, the very low black hole accretion rate makes the intensity of the broad lines very weak and much more difficult to detect relative to all other AGN found at high-z. The fact that out of the three type 1 AGN at $z > 6$ currently found by JADES one is a dormant black hole in a relatively quiescent galaxy, despite all the selection effects against this class of objects, indicates that they must be much more numerous and much more common than actively accreting AGN in star-forming galaxies. We have performed a completeness simulation to infer the ability of the JADES survey in detecting black holes with a given mass and accretion rate, at the same redshift as GN-1001830 (see Methods for details). The results are shown in Fig. 3, in which light background colours indicate higher levels of completeness. As expected, at a given black hole mass, black holes accreting more vigorously are easier to detect. The comparison with the same simulations as in Fig. 2 (green points, from refs. 10,11) shows that GN-1001830 overlaps with the high-mass tail of dormant black holes in the region in which a few of these become detectable in the JADES survey. However, this is just the tip of the iceberg, as most of these dormant black holes are expected to be undetected. Specifically, only 0.1% of the simulated black holes from refs. 10,11 with masses lower than $10^8 M_\odot$ and Eddington ratios below 0.03 are detectable in the JADES survey.

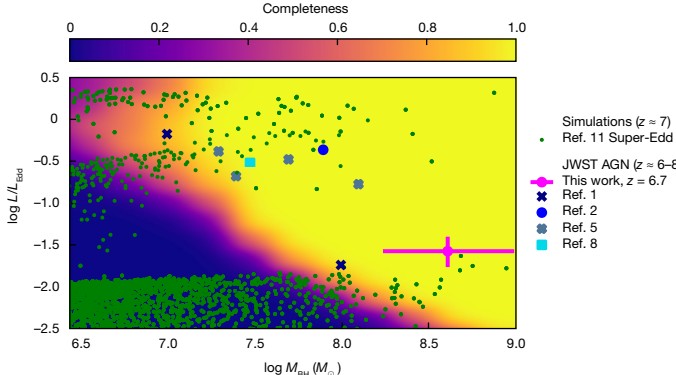

**Fig. 3 | Completeness simulation results on the Eddington ratio versus black hole mass plane.** The blue points show the previously discovered JWST sources at $6 < z < 8$, as in Fig. 2. The dark green points show the simulated AGN (at $z \approx 7$) in the scenario of super-Eddington bursts. GN-1001830 is indicated by a magenta circle with error bars. The colour shading indicates the completeness of the JADES spectroscopic survey in detecting black holes with a given mass and accreting at a given rate relative to Eddington. It can be readily seen that most of the low-accretion rate AGN predicted by super-Eddington bursts lie in the sub-50% completeness region and that GN-1001830 overlaps them at the edge of the high-completeness region. Error bars indicate 1σ uncertainties.

This fraction becomes about 50% in the higher black hole mass range tail probed by GN-1001830 ($10^8 M_\odot < M_{BH} < 10^9 M_\odot$). It should also be noted that the presence of GN-1001830 in the approximately 100 arcmin² of GOODS-N field implies a number density of around $10^{-5.2}$ Mpc⁻³, which is consistent within a factor of two with the prediction of simulations of around $10^{-4.9}$ Mpc⁻³ (ref. 31), especially given that we have not spectroscopically targeted all possible AGN in the fields. The plot also confirms that the several black holes found by JWST accreting close to the Eddington rate (blue symbols) are preferentially selected in this phase only because they are much more luminous and easier to detect. Some of the AGN observed at high redshift are seen accreting at super-Eddington[2,30,37] rates, confirming the existence of (short) super-Eddington phases. GN-1001830 is detected, despite being dormant, because it is just above the detectability threshold; yet, our result indicate that most of the black holes at high redshift are dormant and rare only because they are much more difficult to detect.

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

## Methods

### Data description and reduction

All data used herein have been obtained from the JADES survey, the full description of which is available in ref. 38. The spectroscopic survey consists of several tiers, characterized by their depth, 'Medium' or 'Deep', photometry from which targets were selected, HST or JWST, and the observed field, GOODS-S or GOODS-N. Here we make use of data from the Medium/HST in GOODS-N tier, which consists of a single NIRSpec spectrum in prism and R1000 gratings. We also make use of the accompanying NIRCam wide-band imaging data.

**NIRSpec.** The full description of the NIRSpec data used is available in ref. 2. A summary is provided here for completeness. The observations in the Medium/HST tier in GOODS-N consisted of three medium-resolution gratings (G140M/F070LP, G235M/F170LP and G395M/F290LP) and low-resolution prism. The exposure time was 1.7 h per source in the prism and 0.8 h per source in the medium gratings. The data were processed according to the procedures laid out in ref. 39 and other similar JADES papers, such as ref. 40. A full description of the data reduction procedure will be presented in Carniani, S. et al. (in preparation). Here we note that the spectral data were reduced using the pipeline developed by the NIRSpec GTO team and the ESA NIRSpec Science Operations Team. As the primary interest of this study was the properties of the central, unresolved region containing the AGN, we use the one-dimensional spectra extracted from the central 3 pixels, corresponding to 0.3', of each two-dimensional spectra. Path-loss corrections were calculated for each observation, taking into account the intra-shutter position, assuming a point-source geometry and a 5-pixel extraction box. Owing to the compact nature of the object, to maximize the signal-to-noise ratio (S/N), we used 3-pixel extractions. Although a 3-pixel box is not the extraction box we optimized the path-loss corrections for, we compared directly the two spectra and found no systematic difference within the uncertainties.

The full prism spectrum obtained is shown in Extended Data Fig. 1.

**NIRCam.** The imaging data consisted of seven wide (F090W, F115W, F150W, F200W, F277W, F356W and F444W) and one medium (F410M) filter bands of the NIRCam instrument in the GOODS-N field. We also used imaging in the F182M and F210M medium bands. The photometric data reduction procedure is presented in refs. 41–43 with a full description to be made available in Tacchella et al. (manuscript in preparation). In summary, we used v.1.9.2 of the JWST calibration pipeline[44] together with the CRDS pipeline mapping context 1039. Stages 1 and 2 of the pipeline were run with our own sky-flat provided for the flat-fielding, otherwise keeping to the default parameters. After stage 2, custom procedures were performed to account for $1/f$ noise and subtract scattered light artefacts, 'wisps', along with the large-scale background. Astrometric alignment was performed using a custom version of JWST TweakReg, with corrections derived from HST F814W and F160W mosaics along with GAIA Early Data Release 3 astrometry. The images of individual exposures were then stacked in stage 3 of the pipeline with the final pixel scale being 0.03' per pixel.

### Spectral fitting and further spectral analysis

To identify the broad component in the Hα line, we used a Bayesian method to model it with two components models—one containing only narrow emission in the Hα line, [NII]$\lambda\lambda$6548,6583 and [SII]$\lambda\lambda$6716,6731 doublets, the broad-line model included a broad component in the Hα line. Narrow-line widths were constrained to be the same for every line, the ratio of the [NII] doublet fluxes was fixed to 3 (as from their Einstein coefficients ratios), the [SII] doublet fluxes remained independent but constrained to be within the flux ratio 6,716/6,731 range expected in the low- and high-density regimes (0.45–1.45; ref. 45). The priors on the peak widths were uniform with the fitted full width at half-maximum

(FWHM) ranging between 700 and 1,500 km s$^{-1}$ for the narrow, and between 1,500 and 11,500 km s$^{-1}$ for the broad component, with the lower bound set by instrument resolution. The posterior is estimated with a Markov Chain Monte Carlo integrator[46]. Redshifts for the narrow peaks and the BLR were fit independently with priors being set to narrow Gaussians centred on the overall redshift obtained through visual inspection and widths inferred from the pixel scale in redshift space. Line peak heights used log-uniform priors.

The performance of the two models was quantified using the Bayesian information criterion (BIC), defined as

$$\mathrm{BIC} = \chi^2 + k \ln n, \tag{1}$$

where $k$ is the number of free parameters and $n$ is the number of data points fitted. Following the criteria in ref. 2, we require ΔBIC = $\mathrm{BIC_{Narrow}} - \mathrm{BIC_{Narrow+Broad}}$ to be above 5 for robust detection. Moreover, we require the fitted broad component to have a significance of at least 5$\sigma$. The significance of the broad component of GN-1001830 was found to be 9$\sigma$ and ΔBIC was 31. The full summary of the fit is shown in a corner plot in Supplementary Fig. 2. As shown in this figure, the data are quite constraining on the different components with no significant degeneracies between them.

We note that the spectrum shown in Fig. 1 seems to have a feature at $\lambda \approx 4.94$ μm, the origin of which is currently unclear. In the two-dimensional spectrum, this feature is offset by 1–2 pixels, suggesting that it might be Hα line of a foreground galaxy at $z \approx 6.53$. However, one of the four exposures has an outlier at this location (Extended Data Fig. 2); although this has been masked, the feature might be a residual artefact. For these reasons, we choose to mask this feature in the final fit, although its inclusion would not significantly affect the results. We also note that the Hβ line in Extended Data Fig. 1 seems to contain a similar artefact, which manifests as an apparent broad wing. However, the inferred 2$\sigma$ significance of the feature leads us to conclude that its origin is noise.

To check for evidence of outflows, we fit the Hβ line together with the [OIII] doublet in the medium-resolution data. For this purpose, we fit these lines first with single components constrained to have the same width, with the ratio of [OIII] doublet peaks fixed at 3, then introduce a broader outflow component into each line, and finally, we fit a broad component to Hβ. We find that single narrow component fits are preferred for each line, showing no evidence for outflows or a broad component in Hβ, as can be seen in Extended Data Fig. 3. The measured FWHM of the narrow lines in R1000 was $225^{+11}_{-11}$ km s$^{-1}$, when corrected for instrumental broadening, using the point-source line spread function (LSF) models in ref. 47. This broadening is 180 km s$^{-1}$ in the wavelength range considered.

We also use HST imaging of the source carried out in 2018 to check its variability and thus the presence of supernovae that could potentially produce broad Hα emission while in their nebular phase. Subtracting HST and JWST images taken in equivalent filters shows that the flux of the source varied by no more than 5% over in the 4-year period in the observed frame, corresponding to <5% variability over a 6-month period in the frame of the object. This conclusively shows that our observed broad component cannot be the product of a recent supernova explosion in the galaxy. We also rule out the possibility that the broad-line component may be attributed to the effect of multiple supernovae given the low SFR of the galaxy.

Furthermore, we leverage the higher depth of the prism to fit weaker emission lines, in particular, Hγ, [OIII]$\lambda$4363, [Ne III]$\lambda$3869 and [OII]$\lambda$3727, as the [OIII] auroral line may be used to constrain metallicity, whereas the remaining lines are candidate diagnostics for type 2 AGN. Each line has been fit with a single Gaussian profile together with Hβ and the [OIII] doublet with redshift and FWHM fixed by the latter. We note that, although the [OII]$\lambda$3727 line is part of a doublet, this doublet is completely blended in the prism spectrum and the overall detection

significance ends up being marginal (Extended Data Table 1). All narrow emission lines fitted are summarized in Extended Data Table 1. It should be noted that we carry out our fitting using spectra extracted from the central 3 pixels of the source to enhance the S/N of lines as the region of emission is compact because of the AGN nature of the source. However, the flux corrections applied by the reduction pipeline are geared towards the 5-pixel extracted spectra; we thus redo our fits using the 5-pixel spectra and find that the derived line fluxes differ by less than $1\sigma$.

Our tentative ($3.6\sigma$) detection of [OIII]$\lambda$4363, together with the [OIII]$\lambda\lambda$5007,4959 doublet, can be used to estimate electron temperature based on their ratio. The electron density cannot be reliably estimated from the spectrum because of the lack of an [SII]$\lambda\lambda$6716,6731 detection. However, we note that the inferred electron temperature is relatively insensitive to a density between 100 cm$^{-3}$ and 10,000 cm$^{-3}$, typical of narrow-line regions of AGN[48]. We thus assume electron density of the order of 1,000 cm$^{-3}$ and calculate the electron temperature to be $T_e \approx 25,000^{+3,200}_{-4,800}$ K. Assuming the main ionization mechanism to be AGN activity, we follow the methods in ref. 49 to derive the contributions to metallicity from different ionic species of oxygen— $12 + \log(O^{++}/H) = 7.28^{+0.16}_{-0.1}$ and $12 + \log(O^{+}/H) < 6.13$, the latter value presenting an upper limit due to the low detection significance of [OII]$\lambda$3727. The contribution of higher ionization oxygen species is probably negligible because of the lack of [OIV] and HeII line detections. HeII lines have similar ionization potential to [OIV], but helium is much more abundant; thus, the lack of detection of HeII lines implies that the radiation is not sufficiently hard to produce significant amounts of highly ionized oxygen. The final oxygen abundance ratio estimate is thus $12 + \log(O/H) = 7.32^{+0.16}_{-0.10}$, corresponding to $Z \approx 0.04 Z_\odot$. However, the [OIII]$\lambda$4363 line is heavily blended with H$\gamma$ in prism. Therefore, the simple two-Gaussian fit may underestimate the relevant uncertainties. As a conservative estimate, we obtain a lower limit on the [OIII]$\lambda$4363 flux by fitting the blended lines with a single Gaussian profile and subtracting from its flux the H$\gamma$ flux obtained in a fit without [OIII]$\lambda$4363 included. This method resulted in $F_{[OIII]\lambda4363} \geq 9.1 \times 10^{-20}$ erg s$^{-1}$ cm$^{-2}$. Repeating the former analysis gives $T_e \geq 15,000$ K and $12 + \log(O/H) \leq 7.72$. A lower limit on metallicity was derived by fitting the blended feature with [OIII]$\lambda$4363 only, which results in $T_e \leq 34,000$ K and $12 + \log(O/H) \geq 7.08$. These limits are consistent with the best-fit value and place our source below the mass–metallicity relation[50,51] at similar redshifts (Supplementary Fig. 3).

As a final check on these results, we fit the blended [OIII]$\lambda$4363 and H$\gamma$ feature by fixing the ratio of H$\beta$ and H$\gamma$ to the appropriate Balmer decrement and find $12 + \log(O/H) = 7.25^{+0.20}_{-0.17}$, which is completely consistent with the above estimates.

## Morphological and photometric fitting and stellar population properties

To constrain the stellar mass and SFR of the host galaxy, we use fractional spectral energy distribution (SED) fitting. To do this, we use the tool ForcePho (Johnson, B., manuscript in preparation), which enables us to forward model the light distribution using a combination of Sérsic profiles. We perform spatially resolved photometry with ForcePho following the methodology detailed in refs. 42,43. In short, we model the AGN and host galaxy as a central point-source component and underlying host galaxy component, respectively, and fit the light distribution in the individual exposures of all 10 NIRCam bands simultaneously. This enables us to obtain accurate spatially resolved fluxes and morphological parameters for the galaxy. This approach has been used previously in refs. 2,42,43.

Extended Data Fig. 5 shows the data, residual, model and point-source-subtracted host galaxy for the ForcePho fit. We can see that the galaxy + point-source model has fit the data well without leaving significant residuals and the galaxy component appears bright enough for reliable photometry, with S/N ranging from 6 to 40 across our filters. The resulting best-fit morphological parameters are reported in Extended Data Table 2, which shows that the host galaxy is compact ($R_e = 137 \pm 8$ pc) with a disk-like profile (Sérsic index $n \approx 1$). The quoted statistical-only error on $R_e$ is rather small, considering the marginally resolved nature of the source.

The PSF model that is approximated by ForcePho is based on WebbPSF—incorporating forward modelling of the optics of the telescope, with additional calibration provided by field stars. To investigate the uncertainty coming from the PSF approximation used by ForcePho, we also re-fit the data with a different PSF approximation, which includes charge transfer effects, and obtain a 16% smaller radius. We thus adopt a 16% systematic error floor, which results in a final estimate of $R_e = 137 \pm 23$ pc. The fluxes for the point-source component and host galaxy can be seen in Extended Data Fig. 6. We find that the galaxy component dominated (at the 90% level) in all filters except for the F444W, for which the contribution from the AGN broad H$\alpha$ dominates.

The next stage is to fit this SED to obtain the stellar population properties of the host galaxy. To do this, we use the Bayesian SED fitting code Prospector[20], which uses Flexible Stellar Population Synthesis[52] (FSPS) with MIST isochrones[53], nebular line and continuum emission[54] and a Chabrier[55] initial mass function (IMF). We also use the Bayesian SED fitting code Bagpipes[19], which uses stellar population models in ref. 56, alongside the nebular line and continuum emission[54] with a Kroupa[57] IMF. For both codes, we include a flexible two-component dust model following ref. 58 consisting of a birth cloud component (affecting only light from the birth clouds themselves, for example, stars younger than 10 Myr) and a separate diffuse component (affecting light from all sources). We assume a flexible star formation history (SFH) with a continuity prior[59], in which we fit for the ratio between the six SFH bins. Finally, we exclude the F410M and F356W filters from the fit as those contain the [OIII]$\lambda\lambda$5007,4959 doublet and thus may have been contaminated by AGN ionization. This can be seen in the SED in Extended Data Fig. 6, in which we see the host galaxy has increased flux in the F410M and F356W bands compared with the point source although these bands contain [OIII]$\lambda\lambda$5007,4959. This suggests that emission line flux from the AGN may probably still be contributing to the recovered host galaxy SED in these bands justifying our exclusion of these bands from the SED modelling.

Extended Data Fig. 7 shows the resulting Prospector fit (black) to the observed photometry (yellow) with the $\chi$ values below. These fits yield a stellar mass of $\log(M_*/M_\odot) = 9.00^{+0.27}_{-0.25}$ and $\log(M_*/M_\odot) = 8.71^{+0.24}_{-0.52}$ and instantaneous (within the past 10 Myr) SFR = $1.48^{+0.95}_{-0.42}$ M$_\odot$/yr and SFR = $1.25^{+1.13}_{-0.89}$ $M_\odot$ yr$^{-1}$ for BAGPIPES and Prospector, respectively. These results are consistent within $1\sigma$; thus, we combine the chains given by each code for the final estimate—$\log(M_*/M_\odot) = 8.92^{+0.30}_{-0.31}$ and SFR = $1.38^{+0.92}_{-0.45}$ $M_\odot$ yr$^{-1}$.

We also test the effect of fitting the SED of the combined photometry, that is host galaxy + point source, to estimate how much we would overestimate the stellar mass of the galaxy by not decomposing the AGN and host galaxy components. We find that the results are altered by less than $1\sigma$.

To further test the validity of our decompositions, we perform morphological analysis on the stacked NIRCam images of the source, comparing them with a model PSF. This comparison is carried out in two NIRCam bands—the F115W and F277W. The former was chosen because its PSF is the smallest among the bands not contaminated by the Lyman break, which prevents the use of F090W. The F277W band was chosen as it smears the source over more pixels, better sampling the PSF, whereas our ForcePho fits indicate that the PSF component is still sub-dominant even in this filter.

When analysing the F277W band, we perform an isophotal fit for both the source and the model PSF using the Photutils package[60]. The results of this fit are shown in the top row of Supplementary Fig. 4. As shown in this figure, the best-fit ellipsoidal isophotes for GN-1001830 are

slightly elongated, suggesting some extended morphology, whereas the isophotes of the PSF are circular. The radial profiles (Supplementary Fig. 4, rightmost column) show that our object is more extended than the PSF.

Radial profiles in the F115W band were estimated by placing concentric circular apertures on both our source and the model PSF as the flux was dispersed over too few pixels, undersampling the PSF and making isophotal fits non-viable. Nevertheless, as shown in Supplementary Fig. 4 (bottom right), our source, although compact, is still significantly more extended than the PSF.

All numeric properties of the host galaxy and its black hole obtained from the spectral and SED fitting are summarized in Extended Data Table 2.

Supplementary Fig. 5 shows the SFR of the host galaxy against lookback time (and redshift) presenting the SFH as derived by both Prospector and Bagpipes. Both SFHs are consistent with a flat SFH although the two codes show systematic offsets, probably resulting from different assumptions in the codes, in particular, the different stellar populations used. This suggests that the galaxy experienced almost constant star formation over the past few 100 Myr. A comparison of GN-1001830 with the star-forming main sequence at similar redshifts is shown in Extended Data Fig. 4. Our source lies below the star-forming locus of similarly massive galaxies by a factor of about 3 and would take about 1 Gyr (that is, about the age of the Universe at $z \sim 6$) to double its mass with the current SFR. This indicates that the galaxy is currently fairly quiescent and may have been so for quite some time, although uncertainties on the SFH are large. The presence of AGN activity might offer a possible explanation for this state of the host, suggesting that AGN negative feedback might be responsible for suppressing star formation.

We also fit the extracted point source (AGN) component with a reddened power law with a fixed slope of $\beta = -1.55$ ($F_\lambda \propto \lambda^\beta$), corresponding to an average slope of type 1, unobscured high-$z$ quasars[61]. Assuming the SMC extinction curve, the resulting $A_V$ is $2.68 \pm 1.00$, which, although poorly constrained, is fully consistent with the value derived from spectroscopy. The bolometric luminosity inferred from the fitted $\lambda L_\lambda$ at 5,100 Å, with the bolometric correction from ref. 62, and corrected for absorption, is $4.6^{+7.5}_{-3.2} \times 10^{44}$ erg s$^{-1}$, which is highly uncertain but consistent with the value obtained from the broad component of Hα.

### Estimation of the black hole mass and accretion rate

As mentioned in the main text, the broad component of Hα can be used to infer the black hole mass by assuming the local scaling relations are valid at high redshift and, specifically, from the equation[17,18]:

$$\log \frac{M_{BH}}{M_\odot} = 6.60 + 0.47 \log \left( \frac{L_{H\alpha}}{10^{42} \text{ erg s}^{-1}} \right) + 2.06 \log \left( \frac{FWHM_{H\alpha}}{1,000 \text{ km s}^{-1}} \right) \quad (2)$$

The best fitting values for the broad component of Hα (FWHM of $5,700^{+1,700}_{-1,100}$ km s$^{-1}$ and flux of $27.3^{+4.1}_{-4.0} \times 10^{-19}$ erg s$^{-1}$ cm$^{-2}$) give $\log M_{BH}/M_\odot = 8.23^{+0.38}_{-0.36}$, with the scatter on equation (2) included in the uncertainties. Coupled with the bolometric luminosity of $2 \times 10^{44}$ erg s$^{-1}$, estimated from the broad component of Hα (following the scaling relation given by ref. 63), gives an Eddington ratio $\lambda_{Edd} = 0.009^{+0.005}_{-0.003}$, with a systematic scatter of 0.5 dex.

We note that a recent measurement of the black hole mass in a super-Eddington accreting quasar at $z \sim 2$ has cast doubts on the validity of the ultraviolet (UV) virial relations for single-epoch black hole measurements[30]. The same work, however, points out that when using Hα to measure the black hole mass, the discrepancy is only a factor of 2.5. Moreover, the discrepancy has been ascribed to deviations in the BLR size in the super-Eddington regime, which is certainly not the case for GN-1001830. In summary, the black hole mass measurement inferred from the broad Hα in GN-1001830 is reasonably solid.

It is difficult to estimate dust attenuation in this object. The lack of broad Hβ does not provide strong constraints: even assuming a standard case B recombination ratio of 2.8, the broad Hα line flux implies that the broad Hβ is not detectable. We, therefore, assume, as found for other AGN at high $z$, that the bulk of the obscuration towards the BLR also affects the narrow components[64]. From the observed ratio of the narrow components of Hα and Hβ ($5.51^{+0.86}_{-0.69}$), and assuming an SMC extinction law (appropriate for high-$z$ AGN[65,66]), we infer $A_V = 2.00^{+0.44}_{-0.41}$ mag. We also repeat the estimate using ratios of Hα and Hβ lines to Hγ. This yields $A_V = 2.31^{+0.81}_{-0.61}$ mag and $2.6^{+2.6}_{-2.1}$ mag, respectively. These values are quite uncertain because of the lower brightness of Hγ and its proximity to the Hβ line but remain consistent with the previous estimate. The extinction-corrected black hole mass, bolometric luminosity and Eddington ratio are $\log M_{BH}/M_\odot = 8.61^{+0.38}_{-0.37}$ (with the uncertainty including intrinsic scatter on the virial relation), $L_{bol} = 10^{45}$ erg s$^{-1}$ and $\lambda_{Edd} = 0.024^{+0.011}_{-0.008}$, respectively, with the same intrinsic scatter of 0.5 dex. The extinction correction is uncertain because of the use of the narrow lines. However, the extinction-corrected values of black hole mass and Eddington ratio are still consistent with the uncorrected ones within $2\sigma$. Hence, our conclusions are not significantly altered by the presence of dust. Furthermore, in the next section, we show that fitting the nuclear source detected by NIRCam with a dust-reddened AGN slope results in an extinction consistent with that inferred from the narrow lines. In the previous section, we also infer the bolometric luminosity from the fitting of the nuclear SED and, although with large uncertainties, independently obtain a value consistent with that obtained from the broad component of Hα. As an additional check, we infer the bolometric AGN luminosity from the luminosity of the narrow Hβ and [OIII]$\lambda$5007 lines using the scaling relations in ref. 67. We obtain $L_{bol,H\beta(N)} = 4.7^{+0.7}_{-0.7} \times 10^{44}$ erg s$^{-1}$ and $L_{bol,[OIII]} = 4.9^{+0.1}_{-0.1} \times 10^{44}$ erg s$^{-1}$, which, although lower than the broad Hα estimate, are still consistent with it once the 0.3–0.4 dex scatter on the calibrations is taken into account. Moreover, we use the 5,100 Å luminosity from the previous section to independently infer the black hole mass, which results in $\log M_{BH}/M_\odot = 8.1 \pm 0.8$, with the large error coming from the uncertainties in source decomposition (being the AGN light sub-dominant), reddening and intrinsic scatter on the virial relations. This value is consistent with estimates using broad Hα, but the uncertainty makes it rather unconstraining.

We note that even without correcting for extinction, the resulting lower limit on the black hole mass would still imply a black hole to stellar mass ratio several 100 times above the local relation.

### Black hole scaling relations with $\sigma$ and dynamical mass

As discussed in ref. 2, although high-$z$ AGNs are offset on the black hole–stellar mass plane relative to the local relation, they are much closer to the local relation between black hole mass and stellar velocity dispersion $\sigma_*$ relation and to the local relation between black hole mass and host galaxy dynamical mass. Here we explore the location of GN-1001830 on the latter two scaling relations.

We use the width of the [OIII] line as a proxy of the velocity dispersion, FWHM = $255 \pm 38$ km s$^{-1}$, as measured from the medium-resolution grating spectrum, deconvolved for the LSF for compact sources. The uncertainty in the FWHM value for the narrow lines includes the 20% systematic uncertainty in the LSF broadening, which is not significant for the broad component. We then derive the stellar velocity dispersion by correcting gaseous velocity dispersion by 0.1 dex to obtain the stellar velocity dispersion, following ref. 68, giving $\sigma_* = 121^{+16}_{-16}$ km s$^{-1}$, with 0.3 dex intrinsic scatter for $\log \sigma_* = 2.08 \pm 0.32$. The resulting location of GN-1001830 in Extended Data Fig. 8 (magenta circle) shows that our object is close to the local relation (solid line, with dispersion shown with a shaded region), as are other high-$z$ AGNs previously measured by JWST.

We also use the host galaxy parameters to estimate its dynamical mass using the same approach as in refs. 2,4, which makes use of the equation

$$M_{dyn} = K(n)K(q)\frac{\sigma_*^2 R_e}{G}, \tag{3}$$

where $K(n) = 8.87 - 0.831n + 0.0241n^2$, with Sérsic index $n$, $K(q) = [0.87 + 0.38e^{-3.71(1-q)}]^2$, where $q$ is the axis ratio[69], $R_e$ is the estimated effective radius and $\sigma_*$ is the stellar velocity dispersion. Using equation (3) along with values for $R_e$, $n$ and $q$ given by ForcePho (summarized in Extended Data Table 2) gives $\log M_{dyn}/M_\odot = 9.50^{+0.39}_{-0.39}$, with the uncertainty dominated by the intrinsic scatter on equation (3). We warn that the errors on this estimate are probably underestimated because of the absence of high-resolution spectral observations for GN-1001830. Moreover, the $M_{dyn}$ value may be underestimated as our object is not fully centred in the slit (Supplementary Fig. 1), which may cut off part of the rotation curve in case of source rotation. The position of our object with respect to other JWST sources and the local scaling relation on the $M_{BH}-M_{dyn}$ and $M_{BH}-\log\sigma_*$ star plane is shown in Extended Data Fig. 8. The source remains above the local relation; however, the difference is not as severe as in the black hole–stellar mass relation.

## Gas fraction and SFE
Following our dynamical and stellar mass estimates, we can obtain an estimate for the gas mass in the host galaxy, using $M_{gas} = M_{dyn} - M_*$, assuming little contribution from dark matter, especially at such early epochs, within the central few 100 pc. This gives $\log M_{gas}/M_\odot = 9.37^{+0.39}_{-0.39}$ and a gas fraction $f_{gas}$ of $0.74^{+0.18}_{-0.30}$. The depletion time for our object can thus be estimated as $\frac{M_{gas}}{SFR} = 1.59^{+0.91}_{-0.85}$ Gyr. The depletion time given by the scaling relation from ref. 70, assuming a star-forming main sequence of the form given in ref. 71, evaluates to 0.66 Gyr. Although these values are consistent within the intrinsic scatter on the relation, as shown in Supplementary Fig. 6, this is still suggestive of star formation being inhibited relative to the population of normal star-forming galaxies at this epoch.

## Completeness analysis
To quantify the selection bias affecting the identification of AGN with broad emission lines, we run the fitting procedure described in 1.2 on simulated broad Hα profiles. These profiles were simulated by using the error extension of the prism data for GN-1001830 to simulate Gaussian noise and adding Gaussian line profiles along with a power-law continuum on top of it. The FWHM and luminosity for the narrow Hα component were uniformly sampled from FWHM $\in$ [200, 600] km s$^{-1}$ and $\log[L(\text{erg s}^{-1})] \in$ [42, 43] ranges, respectively. Redshifts of the simulated sources were uniformly drawn from between 6 and 7. Continuum normalization was set to be 100 times smaller than the Hα narrow peak height and slopes were sampled from a uniform distribution ($-1 < \alpha < 1$). The model grid for the broad component of Hα was computed by varying $\log\lambda_{Edd}$ between $-3.0$ and 0.5 in steps of 0.25 and the $\log M_{BH}/M_\odot$ between 5.5 and 9 in steps of 0.25. For bins with $\log M_{BH}/M_\odot < 7$, we performed the simulation assuming the grating spectrum R1000 error extension to assuage the effects of the lower resolution of the prism. Equation (2) and scaling relations from ref. 63 were used to convert the Eddington ratio and black hole mass values to the luminosity and width of the simulated broad component. This yielded a 15 × 15 grid, each point of which contained 100 spectra simulated according to the above recipe.

The fitting of the simulated data was carried out with the same parameters as those described in 1.2, with a bounded least-squares procedure being used to make the fitting computationally tractable. The completeness of each grid point was calculated as the ratio of the sources recovered according to our criteria (in terms of both broad-line significance and ΔBIC) to the sources inserted. The final completeness function is presented in Fig. 3 and shows that we are inherently biased against highly sub-Eddington black holes predicted by simulations, as most of those lie in regions of low completeness. We caution that the procedure described here provides

only completeness with respect to the final step of AGN selection—the fitting of the BLR—and covers only the medium-depth tier of the JADES survey. The full selection function for the JADES survey is more complex as it spans multiple survey tiers, telescope instruments and source selection methods. Its full treatment is thus beyond the scope of the paper, and our results here should be regarded as closer to an upper limit.

## The role of selection effects on the $M_{BH}-M_*$ relation
As discussed in the main text, the finding that most of the black holes at high-$z$ newly discovered by JWST are overmassive on the $M_{BH}-M_*$ could be because of a selection effect, that is, the scatter of the relation is much larger at high redshift and more massive black holes tend to be preferentially selected at high redshift because they are, on average, more luminous. We have shown and confirmed that selection effects play an important part; however, our findings indicate that they cannot completely explain the offset relative to the local $M_{BH}-M_*$ relation.

Specifically, the scenario proposed in ref. 29 (which predicts a black hole–stellar relation similar to the local one but with an order of magnitude scatter) hardly reaches the black hole to stellar mass ratio observed in our object, and their observational bias scenario would require an observed luminosity of about $10^{45-46}$ erg s$^{-1}$, that is, 1–2 dex above the luminosity observed in GN-1001830 before dust obscuration correction (we note that they do not take into account dust extinction, whereas this and most JWST-discovered AGN are affected by extinction).

The Trinity simulation[28] can produce overmassive black holes as observed in GN-1001830, but they require even higher luminosities, in excess of $10^{48}$ erg s$^{-1}$, hence totally incompatible with the luminosity of our object. Therefore, although selection effects are important (as we illustrate more thoroughly in section 'Completeness analysis'), our finding suggests that the overmassive nature of high-$z$ black holes is also associated with an intrinsic offset of the black hole–stellar mass relation, as also suggested by other studies[13].

Finally, our finding that GN-1001830, as well as many other JWST-discovered AGN[2], is closer (or even consistent for many AGN) with the $M_{BH}-\sigma$ and $M_{BH}-M_*$ relations, indicates that the selection effects on black hole mass cannot play a major part, or else the same strong, orders of magnitude, offset should also be present on these relations.

These aspects are, however, outside the scope of this paper and will be discussed more extensively in a dedicated paper.

## Comparison with the FABLE simulations
To further contextualize our findings, we also compare the properties of our source with predictions from the FABLE (Feedback Acting on Baryons in Large-scale Environments)[72] simulations. These are carried out with the massively parallel AREPO code[73] with new comoving 100 h$^{-1}$ Mpc boxes. We consider both Eddington-limited and super-Eddington accretion, bounding our black hole Bondi–Hoyle–Lyttleton accretion rate in two different simulations by 1 and 10× Eddington. Details of other subgrid models, which are largely based on the Illustris galaxy formation models[74], can be found in ref. 72.

A comparison between the simulated sources from FABLE and our object is provided in Extended Data Fig. 9, with the same coding as in Fig. 2, that is, grey symbols show the Eddington-limited scenarios and the red contours the distribution of simulations in which super-Eddington accretion is allowed. As shown in this figure, the super-Eddington simulation can produce more massive black holes with respect to both luminosity and stellar mass in comparison with the Eddington-limited simulation, resulting in a better match with our observations. However, GN-1001830 still lies above our super-Eddington simulations in the right-hand $M_{BH}-M_*$ plot, probably because of the FABLE simulations lacking in volume. To better sample a larger volume, we also include simulation data from magnified simulations of a massive

protocluster ($M_h > 10^{12} M_\odot$ at $z = 6.66$), previously used in ref. 14, which is taken from the larger Millennium simulation volume of 500 $h^{-1}$ Mpc (ref. 75). These points occupy the high-mass region in both panels of Extended Data Fig. 9. The objects obtained in these simulations have stellar masses slightly larger than that inferred from GN-1001830 and, in runs including super-Eddington accretion up to $10\times$ Eddington (and also earlier black hole seeding, unlike the other FABLE results shown here; ref. 14), have more massive black holes. Although a more quantitative match will be explored in future work, these results qualitatively show that super-Eddington bursts more readily explain the properties and presence of objects such as GN-1001830, particularly in increasing the black hole to stellar mass ratio, relative to Eddington-limited scenarios.

We also note that, as with CAT sources discussed in the main text, most of the simulated highly sub-Eddington sources reside in the low completeness region of the JADES survey (Extended Data Fig. 10) and are thus hard to detect at these redshifts even with current instruments.

## Data availability

The reduced data used to make the figures together with the unprocessed data have been made available on the STScI archive as part of JADES Data Release 3.

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

**Acknowledgements** We thank M. Volonteri and A. Fabian for their comments and advice. S.A. acknowledges support from grant PID2021-127718NB-I00 funded by the Spanish Ministry of Science and Innovation/State Agency of Research (MICIN/AEI/10.13039/501100011033). W.M.B. acknowledges support by the Science and Technology Facilities Council (STFC), ERC Advanced Grant 695671 QUENCH. A.J.B. and J.C. acknowledge funding from the FirstGalaxies Advanced Grant from the European Research Council (ERC) under the Horizon 2020 research and innovation programme of the European Union (grant agreement no. 789056). S.Ca. acknowledges support from the HE ERC starting grant no. 101040227—WINGS of the European Union. R.M. acknowledges support from the STFC, the ERC through Advanced Grant 695671 'QUENCH', and the UKRI Frontier Research grant RISEandFALL. R.M. also acknowledges funding from a research professorship from the Royal Society. J.S.B. acknowledges support from the Simons Collaboration on 'Learning the Universe'. D.S. acknowledges support from the STFC. M.P. acknowledges support from the research project PID2021-127718NB-I00 of the Spanish Ministry of Science and Innovation/State Agency of Research (MICIN/AEI/ 10.13039/501100011033) and the Programa Atracciòn de Talento de la Comunidad de Madrid by grant 2018-T2/TIC-11715. B.R. acknowledges support from the NIRCam Science Team contract to the University of Arizona, NAS5-02015 and JWST Program 3215. C.C.W. is supported by NOIRLab, which is managed by the Association of Universities for Research in Astronomy (AURA) under a cooperative agreement with the National Science Foundation. The FABLE simulations were performed on the DiRAC Darwin Supercomputer hosted by the University of Cambridge High Performance Computing Service (http://www.hpc.cam.ac.uk/) provided by Dell using Strategic Research Infrastructure Funding from the Higher Education Funding Council for England and funding from the Science and Technology Facilities Council; the COSMA Data Centric system at Durham University, operated by the Institute for Computational Cosmology on behalf of the STFC DiRAC HPC Facility. This equipment was funded by a BIS National E-infrastructure capital grant ST/K00042X/1, STFC capital grant ST/K00087X/1, DiRAC Operations grant ST/K003267/1 and Durham University. S.K. acknowledges support from St Catharine's College through a Junior Research Fellowship. R.S. acknowledges support from the PRIN 2022 MUR project 2022CB3PJ3—First Light And Galaxy aSsembly (FLAGS) funded by the European Union—Next Generation EU. A.T. and R.V. acknowledge support from the PRIN 2022 MUR project 2022935STW and 2023 INAF Theory Grant 'Theoretical models for Black Holes Archaeology'. M.A.B. acknowledges support from a UKRI Stephen Hawking Fellowship (EP/X04257X/1) as well as from the STFC.

**Author contributions** I.J., R.M. and J.S. contributed to the analysis and initial interpretation of the spectroscopic data. I.J. performed the completeness simulations. All authors contributed to the interpretation of the results. S.A., S. Carniani, M.C., J.W. and M.P. contributed to the NIRSpec data reduction and the development of the NIRSpec pipeline. S.A. contributed to the design and optimization of the MSA configurations. S.T. and W.M.B. contributed to the analysis

and interpretation of the NIRCam imaging data. R.S., A.T. and R.V. contributed to the raw and advanced data products from the CAT simulations. J.S.B., M.A.B., B.J., S.K. and D.S. contributed to the advanced data products from their FABLE simulations. C.D. performed the variability analysis of the source. A.d.G., J.S. and F.D. contributed to the development of tools for spectroscopic data analysis and visualization. B.R. contributed to the JADES data reduction.

**Competing interests** The authors declare no competing interests.

**Additional information**
**Correspondence and requests for materials** should be addressed to Ignas Juodžbalis.

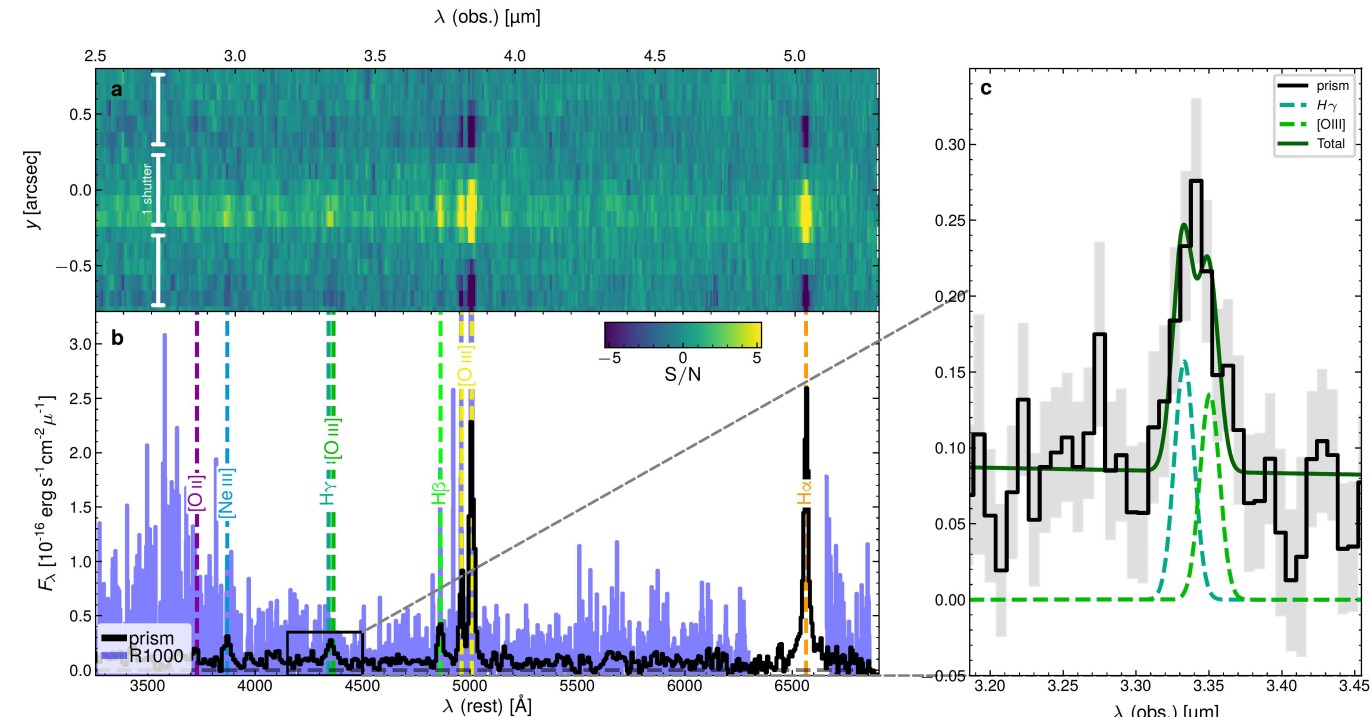

**Extended Data Fig. 1 | Prism spectrum of GN-1001830.** The top panel shows the 2D spectrum with the y axis representing the shutter pitch and yellower portions showing more positive flux. The bottom panel shows the extracted 1D prism spectrum with emission line locations indicated by coloured vertical lines. The (noisier) R1000 spectrum is shown in blue; the wavelength range is narrowed with respect to Fig. 1 to leave out the noisiest parts of R1000. The panel to the right shows a zoomed-in view on the blended Hγ and [O III]λ4363 feature along with its decomposition into two Gaussian profiles. Grey shading indicates 1σ uncertainties.

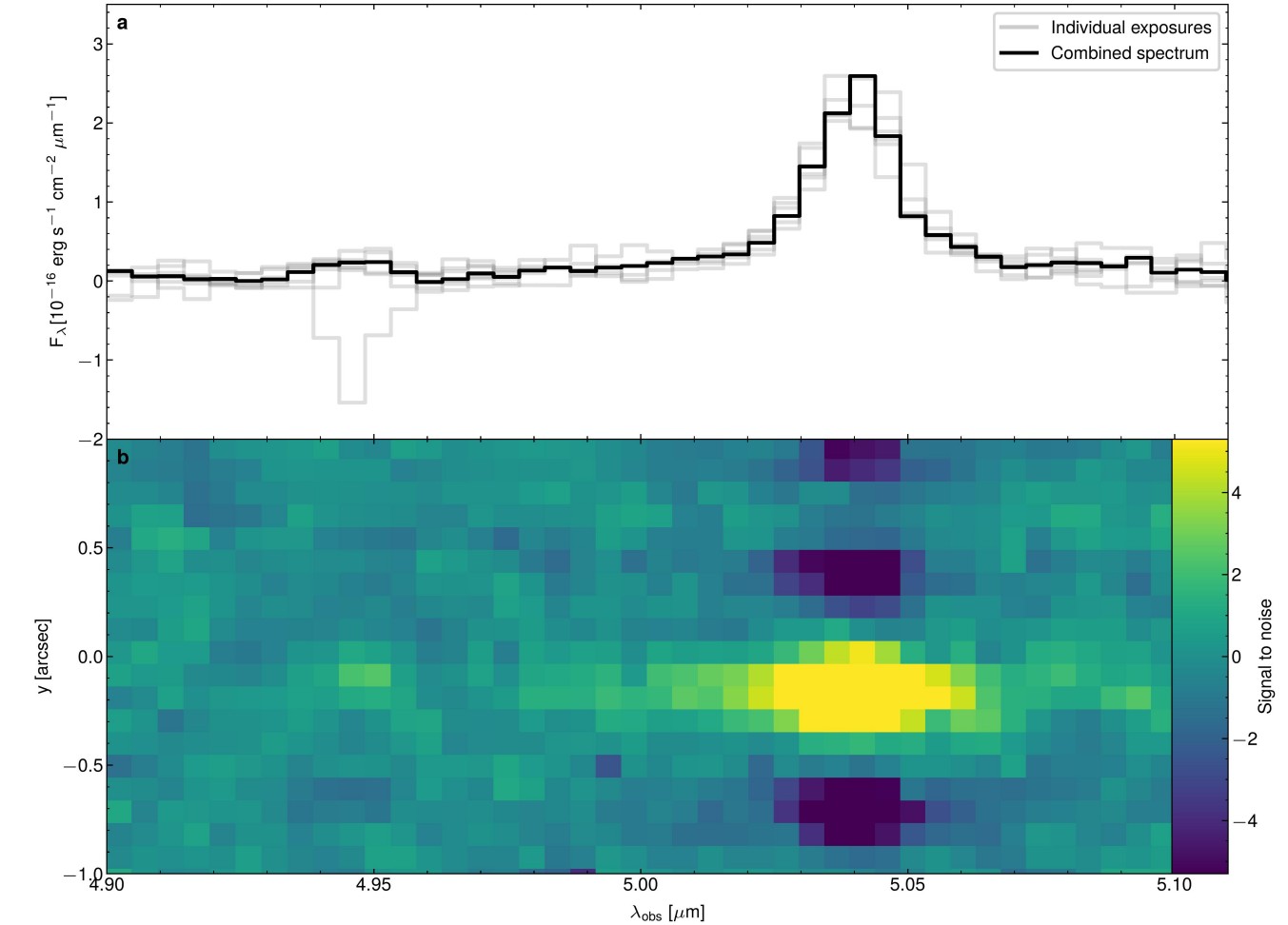

**Extended Data Fig. 2 | Combined spectrum around Hα compared with the individual exposures.** Grey lines in the top panel show the individual exposures, with the stacked spectrum shown in black. The bottom panel shows the 2D spectrum zoomed in on the same region. It can be seen that there is an outlier in the location of the artefact at $\lambda \approx 4.94\,\mu$m. The 2D cutout also shows the slight spatial offset of the feature.

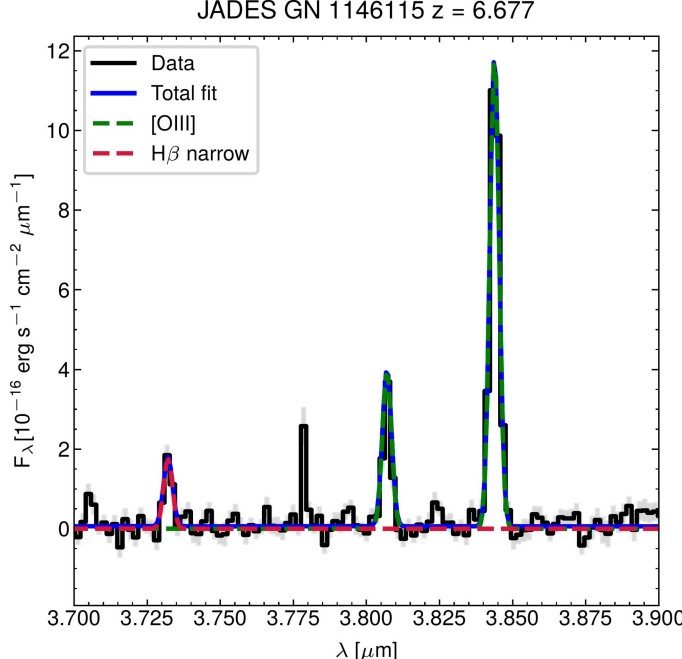

**Extended Data Fig. 3 | Grating spectrum around Hβ and [OIII] along with the best-fit model.** It can be seen that the data are well explained by single component fits to each line, indicating no significant outflows. The spike at 3.775 μm is likely a noise feature that survived sigma clipping. Grey shading indicates 1σ uncertainties.

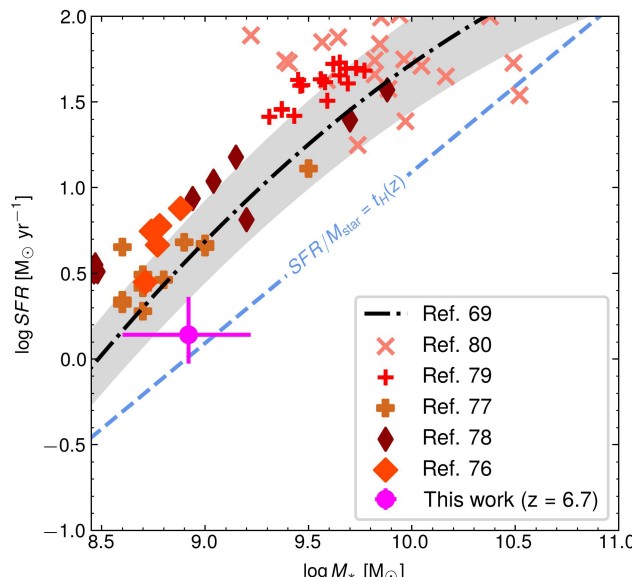

**Extended Data Fig. 4 | Star-forming main sequence.** Star-formation rate versus stellar mass diagram showing the location of GN-1001830 with respect to the star-forming main sequence. The black dash-dotted line shows the star-forming main sequence fit from ref. 71 extrapolated to $z = 6.677$, with the grey shaded area representing the uncertainties. Data at $7 < z < 9$ from refs. 76–79 and ref. 80 are shown with brown and red symbols. The dashed blue line indicates the limit below which it takes a galaxy more than the Hubble time at $z = 6.677$ to double its mass. The magenta circle shows the location of GN-1001830, which is consistent with the dashed line within $1\sigma$. Error bars show $1\sigma$ uncertainties.

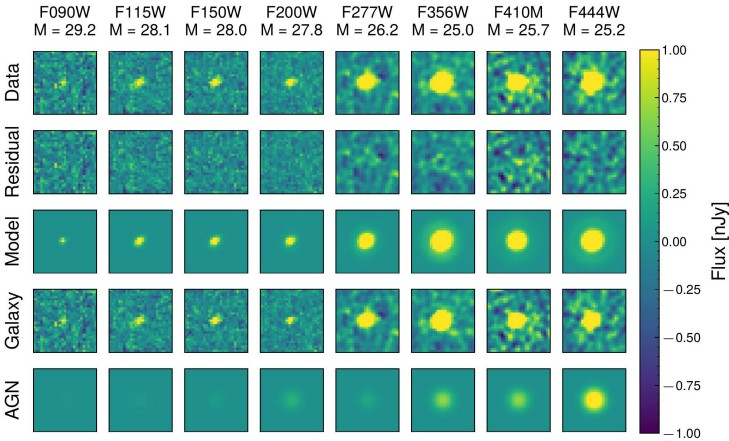

**Extended Data Fig. 5 | AGN-host decompositions.** The data, residual, model, and fluxes with the recovered galaxy component, for the point source and galaxy decomposition in the 8 JADES NIRCam bands. The figure shows that the galaxy+point source model has fit the data well within all bands without leaving significant residuals. The bottom row shows the modelled point source component; stacked magnitudes in each band are shown above each column. Each panel is 0.8 by 0.8 arcsec in size.

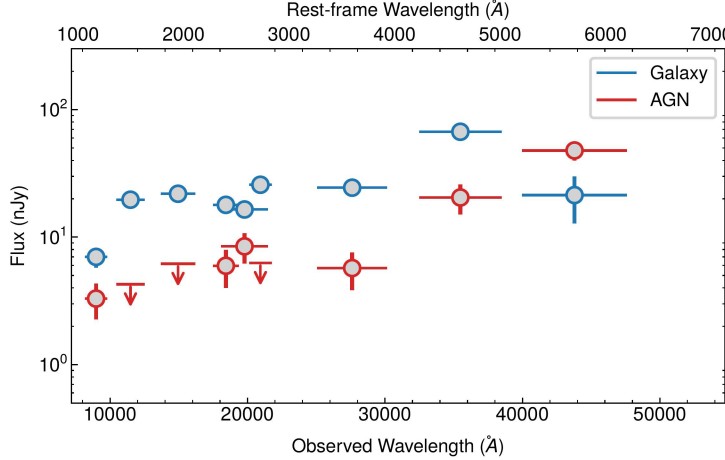

**Extended Data Fig. 6 | AGN-host SEDs.** The spectral energy distribution for the point source and galaxy decomposition in the 10 NIRCam bands. The figure shows that we have recovered a significant amount of the host galaxy's flux within all bands and that the AGN SED is reddened. Horizontal bars indicate filter widths, while vertical ones show $1\sigma$ uncertainties.

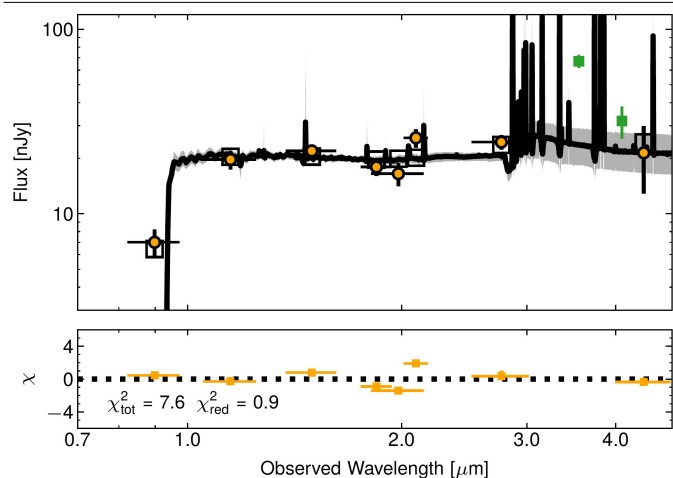

**Extended Data Fig. 7 | Spectral energy distribution (SED) for the host galaxy fit by Prospector.** Yellow points correspond to the observed photometry. Black squares correspond to the model photometry and the model spectrum is overplotted in black. The chi distribution of the observed to model photometry is shown below. We note that we do not fit the F356W and F410M bands, shown in green, due to strong contamination from the AGN which is readily apparent in their excess flux relative to the other bands. The figure shows that Prospector has fit the observed photometry well. Horizontal bars indicate filter widths, while vertical ones show 1$\sigma$ uncertainties.

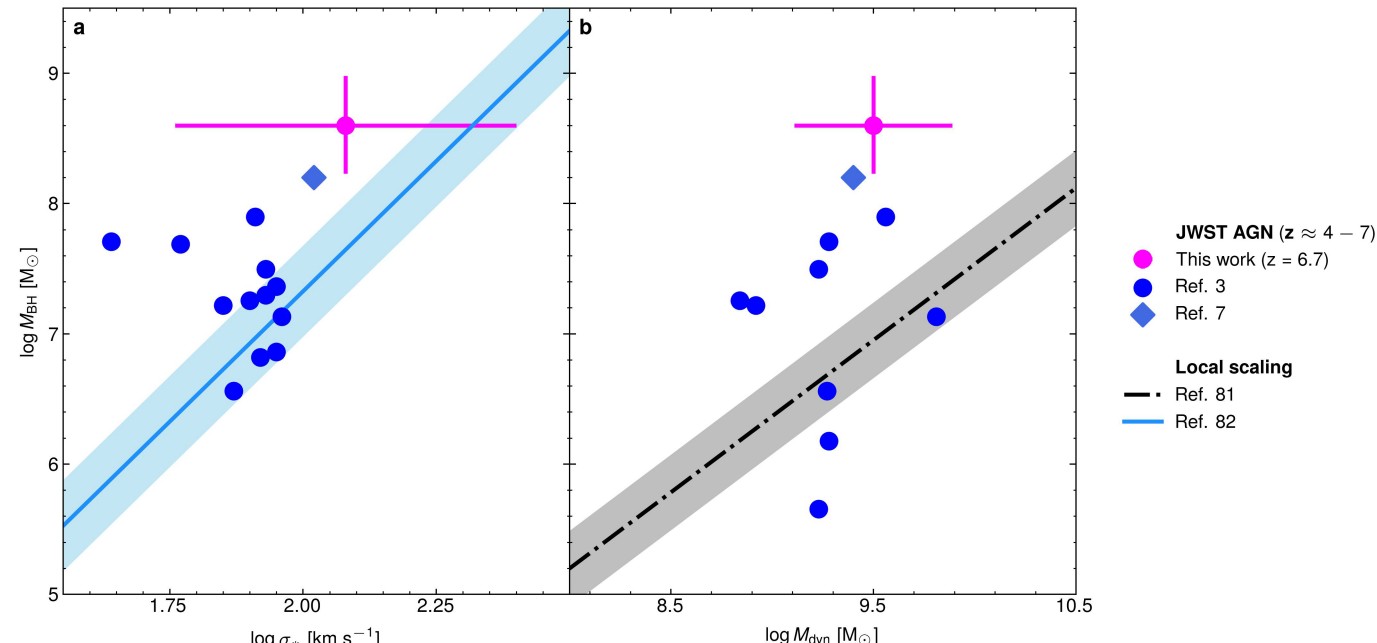

**Extended Data Fig. 8 | Dynamical mass and velocity dispersion comparisons.** Location of GN-1001830 (magenta point) on the black hole mass versus stellar velocity dispersion (left) and versus dynamical mass of the host galaxy (right). Other high-z AGN found by JWST are shown with blue symbols. The black dash dotted line shows the local $M_{BH} - M_{bulge}$ relation from ref. 81. The solid blue line shows the $M_{BH} - \sigma_*$ relation from ref. 82. Shaded areas show the scatter around these relations. While not yet on the local relations, the offset of GN-1001830 is much less severe than in the BH-stellar mass diagram. Error bars show $1\sigma$ uncertainties.

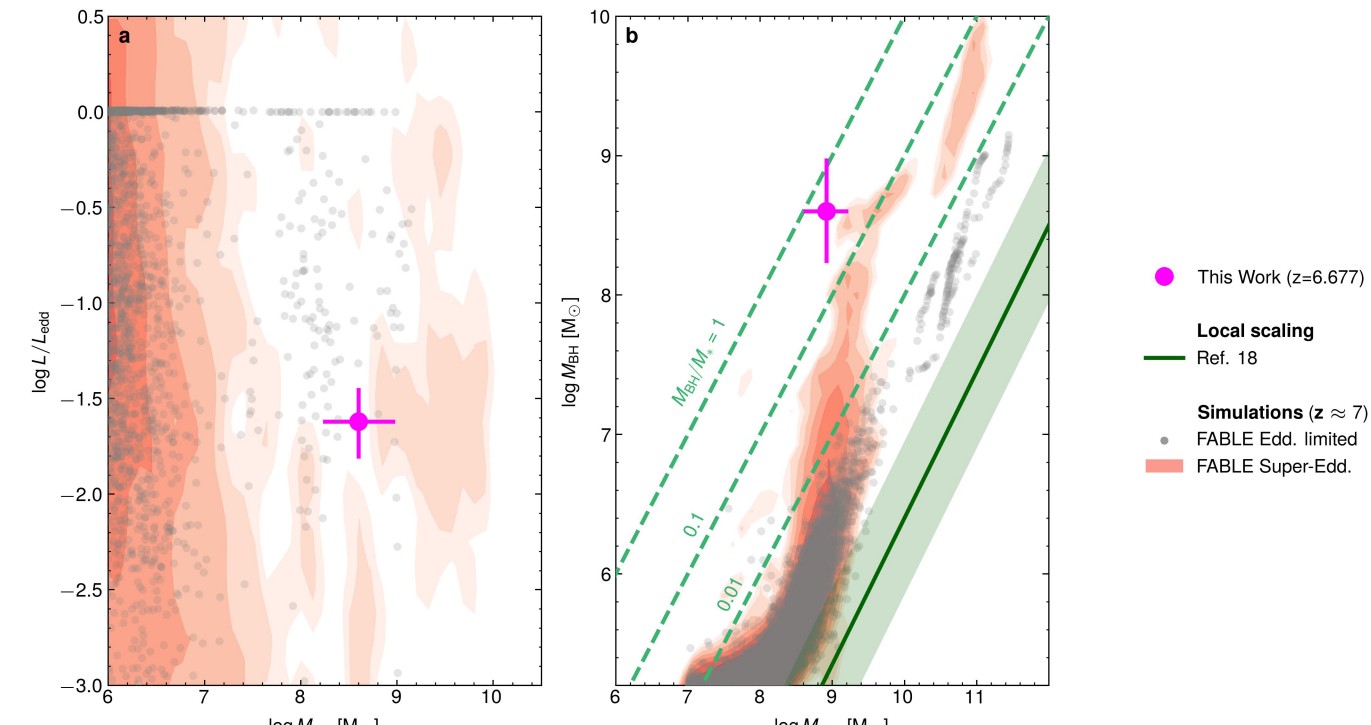

**Extended Data Fig. 9 | Comparison between GN-1001830 and sources in the FABLE simulation.** The layout is the same as the bottom row of Fig. 2, with sources from super-Eddington simulations shown as red contours, while those from sub-Eddington ones are indicated by grey points. As in the case of the CAT models, the Eddington-limited heavy seed scenario fails to simultaneously explain the high BH-to-stellar mass ratio of GN-1001830 and the very low accretion rate. Instead, the scenario in which BHs experience super-Eddington accretion phases can match the properties of GN-1001830, although additional simulations would be need to bridge the gap between the 100 $h^{-1}$ Mpc box and proto-cluster zoom-in simulations. Error bars show 1$\sigma$ uncertainties.

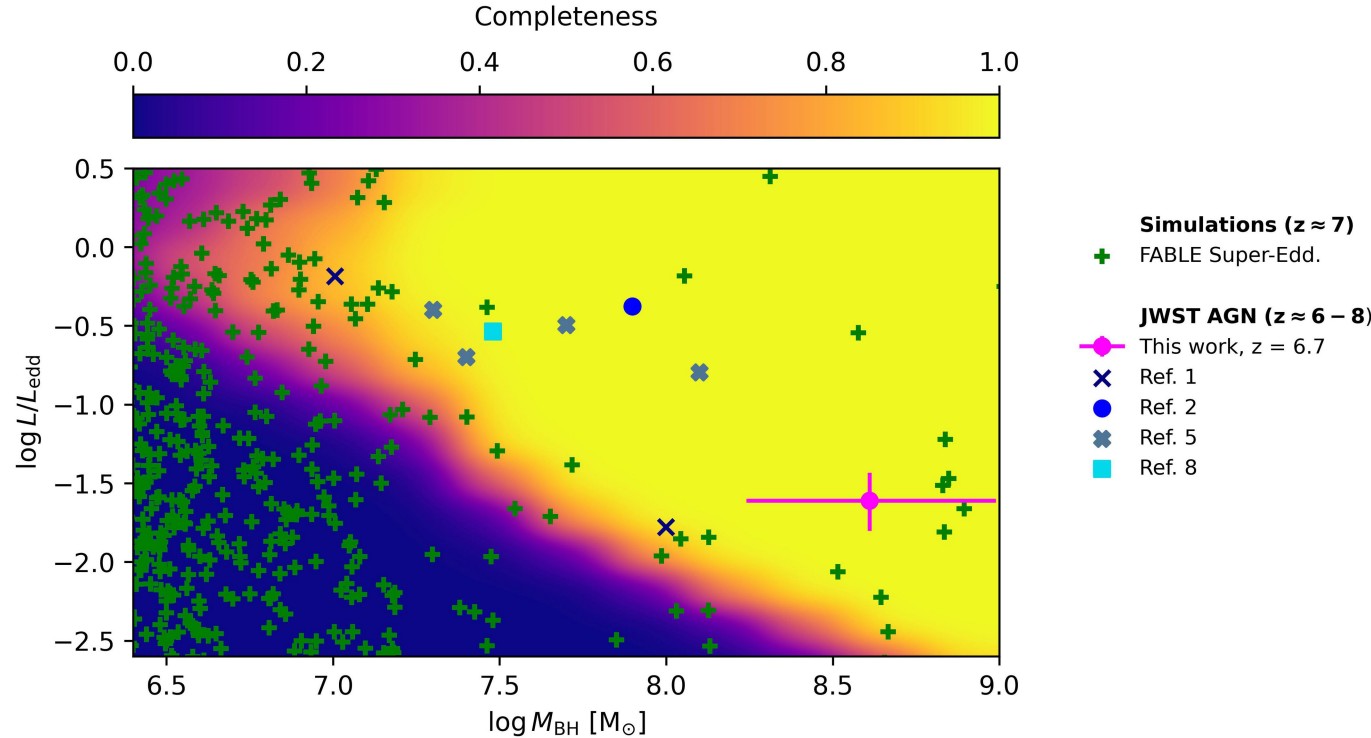

**Extended Data Fig. 10 | Completeness simulations.** Same as Fig. 3, except with super-Eddington models of the FABLE simulation, showing that dormant, post super-Eddington burst AGN are strongly biased against in the current surveys. Error bars show 1$\sigma$ uncertainties.

**Extended Data Table 1 | Summary of all narrow lines measured in GN-1001830**

| Line | Flux ($\times 10^{-19}$ erg s$^{-1}$ cm$^{-2}$) | S/N | Disperser |
|---|---|---|---|
| [S II]$\lambda 6731$ | $< 1.5$ | $1.38\sigma$ | prism |
| [N II]$\lambda 6585$ | $2.84^{+1.06}_{-1.13}$ | $2.0\sigma$ | prism |
| H$\alpha$ | $36.2^{+2.3}_{-2.2}$ | $25\sigma$ | prism |
| [O III]$\lambda 5007$ | $46.7^{+1.3}_{-1.4}$ | $159\sigma$ | R1000 |
| [O III]$\lambda 4959$ | $15.4^{+0.43}_{-0.46}$ | $65\sigma$ | R1000 |
| H$\beta$ | $6.66^{+0.81}_{-0.82}$ | $29\sigma$ | R1000 |
| [O III]$\lambda 4363$ | $2.39^{+0.65}_{-0.65}$ | $3.60\sigma$ | prism |
| H$\gamma$ | $2.77^{+0.68}_{-0.70}$ | $3.96\sigma$ | prism |
| [Ne III]$\lambda 3869$ | $3.87^{+0.94}_{-0.91}$ | $3.63\sigma$ | prism |
| [O II]$\lambda 3727$ | $1.78^{+0.66}_{-0.67}$ | $2.64\sigma$ | prism |

Column one contains the names of each line, the second column measured flux along with uncertainties, the final two columns contain signal to noise ratios and disperser in which each line was measured. For lines with S/N < 3σ the measured fluxes were treated as upper limits; for < 2σ detections, the 2σ upper limit is quoted instead.

**Extended Data Table 2 | Properties of the host galaxy and its AGN obtained via imaging modelling, SED fitting and spectral analysis**

| Quantity | Value |
|---|---|
| $q$ | $0.51^{+0.04}_{-0.04}$ |
| $n$ | $0.94^{+0.07}_{-0.07}$ |
| $R_e$ [pc] | $137^{+23}_{-23}$ |
| $\sigma_*$ [km s$^{-1}$] | $121^{+6}_{-6}$ |
| $\log(M_{\rm dyn}/M_\odot)$ | $9.50^{+0.39}_{-0.39}$ |
| SFR [$M_\odot/yr$] | $1.38^{+0.92}_{-0.45}$ |
| $\log(M_*/M_\odot)$ | $8.92^{+0.30}_{-0.31}$ |
| $\log M_{\rm BH}/M_\odot$ | $8.61^{+0.38}_{-0.37}$ |
| $\lambda_{\rm EDD}$ | $0.024^{+0.011}_{-0.008}$ |
| $12 + \log(O/H)$ | $7.32^{+0.16}_{-0.10}$ |
| $T_e$ [K] | $25,400^{+3,200}_{-4,800}$ |
| $A_V$ | $2.00^{+0.44}_{-0.41}$ |