## [Peer Review File · Nature]

A dormant overmassive black hole in the early Universe

Corresponding Author: Mr Ignas Juodzbalis

Version 0:

Reviewer comments:

Referee #1

(Remarks to the Author)

Dear Authors,

I have reviewed your manuscript titled "A dormant, overmassive black hole in the early Universe". The paper describes a JWST/NIRSpec detection of a very massive, yet largely dormant black hole at $z \sim 6.7$. The combination of having well detected and resolved lines, in addition to an unambiguous presence of the host galaxy, with secure M^* , allows to reliably constrain a vast array of BH and galaxy properties. Moreover, the paper presents an extremely high bh to host mass ratio in a massive BH at high- z , which is extremely relevant and provides another important piece of the puzzle regarding BH growth. Unavailable with such precision in previous JWST studies on the topic. This work, in my opinion, is suitable for a publication in Nature.

However, before I can make a final recommendation, I would like the authors to consider the following comments.

The version of the manuscript I received does not contain line numbers, but I will do my best to point towards relevant pieces in the text.

Major:

1) Page 3: "taking into account the effect of some dust obscuration".

This is a bit nebulous. Mention in one or two words the method used (decrement) and the assumed A_V . Potentially it can also be beneficial to say that the answer you get from SED fitting is also consistent.

2) P5: "This indicates that the galaxy is currently fairly quiescent and may have been so for quite some time, although uncertainties on the SFH are large."

Curious. I believe that following Rawlings & Saunders+91 (also see Marchesini+04) it is possible to derive the Lbol from the narrow OIII line (assuming it comes from NLR).

Given that you already have Lbol measurement from broad Ha (and SED fitting), I wonder how different these estimates would be. If the Lbol from OIII is larger than that from broad Ha, would it be possible to attribute the "extra" emission from narrow OIII to the galaxy instead and re-examine the SFR for example? Would be curious to see if galaxy can even contribute anything there or not.

3) P7: "The Trinity simulation..."

In this work the authors entertain the idea of BH growth through cycles of super-Eddington episodes, followed by periods of relative dormancy. I am curious to what extent Trinity (or any other) simulations can actually model something like that?

I suppose it all comes down to switching on the feedback, but it is unclear to me as to what can regulate the frequency of

these episodes.

4) P7: "we derive a stellar velocity dispersion". This assumes that all OIII is coming from galaxy rather than the NLR? I suppose given the rather faint nature of the AGN it is reasonable. Although I would discuss this assumption in a bit more detail.

5) P8 and throughout.

I see how the argument can be made for episodic accretion and do not aim to lessen its potential impact. Other works on similarly overmassive BH at high-z, e.g. Larson+23 and Kokorev+23 also entertain this possibility as their black holes also can not be reproduced by Eddington limited accretion in some cases.

My gripe with this entire line of thinking however, is the episodic accretion scenario seems somewhat ad hoc. Depending on the starting mass and the number of these super Eddington episodes one can reproduce quite a wide array of black-hole masses at whatever redshift.

This by no means implies that the authors should not entertain this idea, I think it is entirely valid. I just wish we could find a somewhat less "hand wavy" way of explaining these massive black holes so early on.

One other interesting and equally as degenerate and ad hoc scenario is to think about spin and radiative efficiency.

The smaller the spin of the black hole, the more it is able to accrete as the radiative efficiency is also low. For example it is possible to make a black hole from a pop III seed ($\sim 10^2$ Msol) at lets say $z \sim 15$, grow it at almost zero spin to 10^6 at $z \sim 10$, then allow more spin, slow down the accretion and still grow to 10^8 by $z \sim 7-8$. All while staying sub-Eddington, without any bursts/feedback required.

Myself, I have no answer for this. Just something to think about.

6) P9: "the majority of these dormant black holes are expected to be undetected"

Interesting point. Considering the volume sampled by the JADES data, what are the chances to find such a massive BH?

While no real BH mass functions from JWST data yet exist, some efforts were made by Matthee+23 and Kokorev+24, albeit both very uncertain. Also, Kokorev+24 and Dayal+24 discuss the potential for a large number of dormant BHs present at the massive end of the mass function.

I would also look at He+19 mass function albeit it is at lower-z, but could be useful to think about the number densities.

7) Figure 5: Zooming in on the Hbeta line I noticed that it does not look symmetric, with a broader wing on its right side. Have the authors entertained why that might happen?

8) I think somewhere in the data section, both PRISM and medium res spectra should be shown alongside one another.

9) P16: "We therefore assume, as found for other AGN at high-z, that the bulk of the obscuration towards the Broad Line Region also affects the narrow components"

Fair enough. I understand how it can be tough to measure dust extinction in these cases. As much as it is unclear to me if narrow lines can be used instead of broad lines here (and if narrow lines are NLR or galactic), I think the assumptions in this work are clearly stated.

What I propose the authors to carry out, is to see if this A_v estimate holds for the other (narrow) line ratios, like Hb/Hg and Ha/Hg. While Hg is blended with the OIII line checking whether the dust extinction you get from a different ratio of Balmer lines holds, would give more confidence on this estimate.

It is reassuring to see that the Ha/Hb dust extinction is consistent with the one from the optical continuum slope.

Minor:

1) The beautiful spectrum authors show in Figure 5 is one of the main characters of this story. I feel it would be quite nice to show it in the main text instead. I leave this completely up to the authors however.

2) Page 3: Please write the uncertainty on z_{spec} here.

3) P3: "However, the resolution at this wavelength ($R \sim 460$ for a compact source)". Can the authors quote the rest-frame velocity resolution here too.

4) Same line: What is the scaling factor assumed here? I believe it is generally assumed to be roughly ~ 1.3 for point-like

sources. Please reference the relevant work (or works) where this was first addressed (i.e. de Graaff+23).

5) P3: "(whose narrow component is stronger than H α)". The narrow part of Ha presumably? Please clarify.

6) P3: "with a systematic scatter of 0.33 dex". I believe "systematic" and "scatter" are two mutually exclusive terms. What I believe the authors mean here is that they take into account the scatter of both their data and the relation used to obtain Mbh. Very nice that you consider and acknowledge this scatter regardless.

This also appears in a few more sentences throughout the manuscript. Double check please.

7) P3: "reasonably solid" I would change "reasonably solid" to something that inspires a bit more confidence.

Also is it possible to fit the spectrum together with the continuum to try and infer the black hole mass from the underlying 5100 A continuum? Although if the BH is only 2 % Eddington I would expect the contribution to the continuum to be quite negligible.

8) P3: ForcePho reference

9) P4: "The 8 bands photometry", bands-> band

10) P4: "These fits, averaged together," Does it mean that they show more or less consistent results?

11) P4: "instantaneous star formation rate", how instantaneous? 10 Myr or so I suppose.

12) P6: "note that our finding of such". "Discovery" would sound better in this context.

13) P7: "(as for other high-z AGN)"-> as compared to other high-z AGN?

14) P7: "massive seeds". a few more words saying explicitly what DCBH entails could be useful. Something about clouds of pristine gas facilitating direct collapse.

15) P10: "AGN observed at high redshift are seen accreting at super-Eddington". Add Fujimoto+22 (z~7.2) to that list.

16) P10: I am not sure what the authors mean by "tier" here.

17) Sect 1.1.1. Very first line. "At" -> "in"

18) Same paragraph. "The data was processed"-> the data were processed.

19) Fig 9: What is the feature at 3.775 um?

20) P17: "galaxy component appears bright enough for reliable photometry."
Please be more specific here. Quote average S/N per pixel/resolution element? Or just the average S/N in your filters.

21) P17: "illustrates" -> illustrate

22) P17: "galaxy contains significantly more flux". I would rephrase this slightly to read a bit better. Something like "galaxy component is dominated by blue light...". Saying this in this particular way also may start thinking about the blue component found in LRDs and its uncertain (as of yet) origin.

23) Fig 11: Can you add another row showing the AGN model by itself?

24) Fig 11: Consider showing the observed source magnitudes and the uncertainty for the top panels here.

25) Fig 13: The dynamic range of the plot is a bit too wide to see how well the points fit here. As the lines are irrelevant here I suggest to the authors to narrow it down a bit, it's fine to cut off the lines.

26) Fig 13: Can you show the reduced chi2 here in addition or instead of the total one?

Referee #2

(Remarks to the Author)

We were given the paper "A dormant, overmassive black hole in the early universe" by Ignas Juodžbalis et al (ms 2024-03-04689) to referee. This is another groundbreaking result from the JADES team, and we find it both interesting and relevant to the current state of the field. The paper describes an AGN and associated host galaxy at high redshift (z = 6.677) in the JADES GOODS-N field. The system appears to be quiescent with a highly overmassive central black hole in relation to its host stellar mass, and the paper claims that this is a SMBH in a period of quiescence between super-Eddington accretion bursts, embedded in a host galaxy that itself is in a quiescent state (i.e., low star formation). The paper is well-written, and makes a convincing case that there really is a low-luminosity quasar embedded in this galaxy. However, we have some

concerns regarding the analysis of the images and the separation of host galaxy from the central quasar that are core to the analysis, and that we feel need to be addressed before publication. Our comments and suggestions are as follows, starting with our biggest concerns.

- Figure 11 shows the JWST image of this object in multiple filters, and illustrates the process of separating the central point source and the extended galaxy. Doing this well (especially given the very small scale size inferred for the host galaxy; see comments below) is quite tricky, and we are concerned that the systematic errors in the process haven't been fully incorporated into the analysis. In particular, small errors in the assumed point spread function (PSF) have the potential to significantly affect the inferred host galaxy properties. The paper needs to describe how the PSF was actually determined, and the robustness of the results to reasonable variations in the PSF in each of the filters. See for example the discussion in Ding et al (2023; reference 34 in the present paper). It is stated at the top of Page 19 that the stellar mass is altered by less than 1 sigma if "no" PSF subtraction is done. Is this equivalent to saying that the quasar component is close to negligible in the imaging?

- Related to the previous point, the host galaxy stellar mass, size, and star formation rate (which lead to many of the key results of the paper: the large ratio of black hole to stellar mass and the quiescence of the host) all rest heavily on the correct decomposition of quasar and host galaxy light. A more thorough discussion of how you attribute continuum flux to the quasar or galaxy, and the systematic uncertainties in that process, would be appropriate. That is, we are somewhat skeptical of the robustness of the SED fitting given the potential for significant systematic errors on the host photometry. On a related note, the SED of the host and quasar drawn from the imaging (Figures 12 and 13) should be consistent with the shape of the spectrum itself (Figure 5); is this indeed the case? The spectrum has a blue continuum (at least in f_{λ}), while the SED is quite red (in f_{ν}); are they consistent? Showing that the spectra and SEDs match would go a long way in confirming the robustness of the photometry, although it is not clear if it will give further insights into the quasar/host decomposition.

- Table 2 lists the fit parameters from the imaging and spectroscopy. The effective radius of the host galaxy is given as 137 ± 8 parsecs. At this redshift, this corresponds to $0.025''$, while the diffraction limit at 356W (where Figure 5 implies the host galaxy to be well-defined) is about $0.1''$. It is of course possible to measure scale sizes significantly smaller than the PSF with sufficiently high S/N data and a sufficiently accurate model of the PSF, but we would like to be convinced that this measurement is robust. Note that this is true even if the quasar contribution to the image is negligible, as the galaxy image itself is also broadened by the PSF. We are similarly suspicious of the tiny error bar on the scale size (and the similarly small error bar on the Sersic index n).

- Similarly, the dynamical mass of the galaxy depends on R_e , as well as σ_* , measured from the width of the [OIII] line (Figure 9). The line appears to have a full width at half maximum of about 3 pixels, i.e., it is very close to being unresolved. However, there is no discussion of the uncertainty in the value of σ_* due to uncertainties in the resolution of the spectrum, or the fact that the inferred velocity dispersion is in effect the difference in quadrature of the measured line width and the resolution of the spectrograph. The quoted uncertainty on σ_* is 6 km/s, which seems unrealistically small. This leads to an extremely small uncertainty on the dynamical mass of the galaxy, 0.04 dex; we are not at all convinced of this.

- The black hole mass estimate is based on a calibration from low redshift; it should be stated explicitly that one has to assume that this calibration holds at this high redshift. In addition, the low-redshift calibration itself is known to have an intrinsic scatter of order 0.3 dex, which should be included in the error budget.

These are our major concerns. The following are somewhat smaller issues in the paper, that we encourage the authors to consider. Reading sequentially through the paper:

- Page 6: You state that the existence of this source is "incompatible" with the selection effect scenarios. This seems like a strange way to phrase it; perhaps the statement is that the data are deep enough to be less sensitive to selection effects (as you show in Figures 4 and 18).

- The upper left panel of Figure 5 seems to show that [OIII] and H alpha are extended. Is this indeed the case? If so, one can get a measure of star formation rate in the host galaxy which is independent of that determined from the SED fitting.

- Figure 6 shows that the object was not properly centered in the slit. If the host galaxy is rotating, the width of the [OIII] line used for the dynamical mass estimate will be underestimated, as part of the rotation curve is systematically cut off. This is worth some discussion.

- This paper has a total of 18 figures, some of which seem less necessary to show. We suggest dropping Figure 7; Figure 18 is quite similar to Figure 4, and perhaps is not needed either.

- Page 13: You explore the possibility that the object includes a supernova. You should add a brief description of why this possibility is being considered; is the idea that the broad H alpha could be due to a supernova in a nebular phase?

- Page 16: The best-fit extinction is $A_V = 2.0 \pm 0.4$ magnitudes. But you say that the bolometric luminosity changes by less than 2 sigma if you ignore extinction. That's puzzling, as the extinction seems to be detected at 5 sigma (and bolometric luminosity should scale directly with the extinction). This requires a bit more explanation.

- Figure 9 shows a piece of one of the three medium-resolution grating observations of this source. Did the other two include any emission lines?

Note that the y-axis units in Figures 9 and 5 are the same, but labeled differently (10^{-16} erg/s/cm²/micron vs. 10^{-20}

erg/s/cm²/A). However, the strength of the spectral features in the two spectra do not seem to be consistent with each other. This should be made consistent.

-The caption to Figure 11 should state what the spatial scale of each panel is.

-It would be appropriate to include the F356W and F410M bands in Figure 13 (perhaps with a different symbol) even if they are not used in the fit. It would be interesting to see if the effect of the emission lines is directly seen in the photometry.

Page 21, Section 2: given the large inferred gas mass, it would be interesting to speculate on whether molecular gas might be detectable in this source with ALMA.

Table 2: We have already commented on the surprisingly small error bars of many of the tabulated quantities. We also point out that the value of T_e is written with considerably too many significant figures; perhaps it should be, 25400+3200-4800 K.

We would like to see a revised version of the paper that responds to our concerns.

Reviewed by Camryn Phillips and Michael Strauss, Princeton University.

Version 1:

Reviewer comments:

Referee #1

(Remarks to the Author)

Dear Ignas and Co-authors,

Thank you very much for investing your time to carefully consider and address all of the comments.

I have gone through the updated text and figures and do not have any further questions or serious concerns. I did notice a few small things that can be addressed, these are given below. Apart from that, I would like to congratulate the authors on this great work and I am now happy to recommend this paper for a publication in Nature.

#=====

A few small things.

1) Your last bold text on Page 5 says:

"These predict that either black holes are born from relatively massive seeds (e.g. direct collapse black hole, originating from clouds of pristine gas) accreting below the Eddington rate, or from either light seeds (stellar remnants) or heavy experiencing short phases of super-Eddington accretion ([11–17, 32–35]."

I think the "or heavy" part here is redundant since the DCBH are already the heavy seeds.

2) I think it would be better to replace R1000 in Table 1 to G395M. Same in the caption of Fig.4 and the legend of the figure. Up to you though.

3) Section 1.1.1. When you list dispersers and filters as "(G140M/F070LP, G235M/F170LP, G395/F290LP)", the G395M is missing an M.

Referee #2

(Remarks to the Author)

We appreciate the effort the authors have put into addressing our and the other reviewer's concerns, and we find the revised manuscript to be an interesting and thorough work with relevance to the current field. We think the paper should be accepted for publication with only minor updates and an additional editing pass. We do have some remaining concerns, but they are minor enough that there is no need for us to see the paper again before it is accepted.

Reading sequentially through the paper:

Page 6: The paper could use another editing pass to tighten up the

discussion and remove repetition. This was particularly apparent on Page 6, where the addition of several sentences to respond to the referees' concerns makes the discussion choppy and repetitive. For example, super-Eddington accretion is mentioned in three paragraphs in a row, each time making essentially the same point. And the very long paragraph starting, "Finally, we argue that dormant..." also needs tightening.

Figure 4 (page 9): The figure caption should explain the relationship between this figure and the similar Figure 1.

Page 10 is another example of a repetitive and overly lengthy discussion about the presence of a broad component in H beta.

Page 14 describes the PSF modeling and the uncertainties thereof. In the response to our report, the authors stated that the PSF is modeled with a Gaussian Mixture Model, but it is not stated either in the response or the revised paper to which data this model was fit. Stars in the field of view? An a priori model based on the optics of JWST? This should be clarified. In addition, it is stated that a different PSF model which incorporates charge transfer effects allows one to understand the systematic error in the size of the host galaxy. Without knowing more about the PSF modeling process, the reader cannot assess whether this is a full accounting of the PSF uncertainty.

We appreciate the addition of Figure 10 and its demonstration of the extent of the galaxy!

Page 16: It is stated that the F277W band "smears the source over more pixels". Is the point simply that the PSF is largest in this band? In the next paragraph, in F115W "the flux was dispersed over too few pixels". Is this a statement that the PSF is somewhat undersampled?

We have a general concern that the paper may be somewhat too long. This will be a question for the editor to address, but tightening the text in various places (some of which are indicated above) will help. Other areas which felt overly long were:

- The discussion of the extinction on the host galaxy. Multiple methods are used to estimate the extinction. All have very large error bars, and it is not surprising that within these uncertainties, they agree with one another. Quite a bit of text is devoted to saying this; it could be stated more succinctly.
- Section 5 on the future evolution of this system: this seems like pure speculation that doesn't directly support anything in the main paper, and thus seems out of place in the supplementary materials section of the paper.

We would like to thank the referees for providing many useful and insightful comments, which we address in a point-by-point response below.

Referee #1

Major:

1) Page 3: "taking into account the effect of some dust obscuration".

This is a bit nebulous. Mention in one or two words the method used (decrement) and the assumed A_v . Potentially it can also be beneficial to say that the answer you get from SED fitting is also consistent.

We have included an additional clarifying statement in that sentence. However, due to editorial concerns most of the extended discussion on extinction is presented in Methods (Sect. 1.4).

2) P5: "This indicates that the galaxy is currently fairly quiescent and may have been so for quite some time, although uncertainties on the SFH are large."

Curious. I believe that following Rawlings & Saunders+91 (also see Marchesini+04) it is possible to derive the L_{bol} from the narrow OIII line (assuming it comes from NLR).

Given that you already have L_{bol} measurement from broad Ha (and SED fitting), I wonder how different these estimates would be. If the L_{bol} from OIII is larger than that from broad Ha, would it be possible to attribute the "extra" emission from narrow OIII to the galaxy instead and re-examine the SFR for example? Would be curious to see if galaxy can even contribute anything there or not.

In the revised version we perform these estimates and find that the bolometric luminosities obtained from narrow Hbeta and [OIII] lines lie slightly below the estimate from the broad Halpha, however, are consistent with it within 1 sigma of the calibration scatter. A short paragraph discussing this has been added to the methods.

3) P7: "The Trinity simulation..."

In this work the authors entertain the idea of BH growth through cycles of super-Eddington episodes, followed by periods of relative dormancy. I am curious to what extent Trinity (or any other) simulations can actually model something like that?

I suppose it all comes down to switching on the feedback, but it is unclear to me as to what can regulate the frequency of these episodes.

In the case of CAT simulations, which we make the most use of, the super-Eddington bursts are triggered by major galaxy mergers, thus not being entirely ad hoc. Their model predictions were made before the dormant BH that's the subject of the paper was discovered (Trinca+22, Schneider+23).

The FABLE simulations considered likewise model their BH growth in a self-consistent manner in allowing super-Eddington accretion by changing the allowed upper bound on the Bondi-Hoyle-Lyttleton accretion rate prescription. These were also developed before the discovery of our dormant black hole (Koudmani+23, Bennett+24).

As for the Trinity simulations, we are not too familiar with them, but those are not used for comparison in our paper. The Trinity simulations are only discussed in the context of the claimed bias in the MBH-Mstar relation, which is in contrast with our findings.

4) P7: "we derive a stellar velocity dispersion". This assumes that all OIII is coming from galaxy rather than the NLR? I suppose given the rather faint nature of the AGN it is reasonable. Although I would discuss this assumption in a bit more detail.

We cannot really disentangle NLR from ISM in the host galaxy (possibly ionized by star formation), and at this stage we have to live with this uncertainty, although future IFS observations will hopefully clarify this. However, this is not a crucial part of the paper and we can leave it out from the paper if the referee feels strongly about it.

5) P8 and throughout.

I see how the argument can be made for episodic accretion and do not aim to lessen its potential impact. Other works on similarly overmassive BH at high-z, e.g. Larson+23 and Kokorev+23 also entertain this possibility as their black holes also can not be reproduced by Eddington limited accretion in some cases.

My gripe with this entire line of thinking however, is the episodic accretion scenario seems somewhat ad hoc. Depending on the starting mass and the number of these super Eddington episodes one can reproduce quite a wide array of black-hole masses at whatever redshift.

This by no means implies that the authors should not entertain this idea, I think it is entirely valid. I just wish we could find a somewhat less "hand wavy" way of explaining these massive black holes so early on.

One other interesting and equally as degenerate and ad hoc scenario is to think about spin and radiative efficiency.

The smaller the spin of the black hole, the more it is able to accrete as the radiative efficiency is also low. For example it is possible to make a black hole from a pop III seed ($\sim 10^2$ Msol) at lets say $z \sim 15$, grow it at almost zero spin to 10^6 at $z \sim 10$, then allow more spin, slow down the accretion and still grow to 10^8 by $z \sim 7-8$. All while staying sub-Eddington, without any bursts/feedback required.

Myself, I have no answer for this. Just something to think about.

Previous works finding overmassive black holes have indeed invoked other scenarios, primarily heavy seeds (which is a scenario that we consider too). However, in past works they have not tried to match simultaneously both the BH/stellar mass relation *and* the accretion rate; additionally, they were considering primarily AGN accreting closer to the Eddington limit and not so extreme in terms of BH/stellar mass ratio. The novelty of our work is that we draw constraints by both considering the BH/stellar mass ratio *and* the accretion rate, and on a very specific target that is extreme in both terms, and therefore much more constraining. The implementation of the super-Eddington burst is not random, but linked to specific episodes of the evolutionary steps of the galaxy (merging events). Although, the lifting of the Eddington limit is indeed an assumption, that is not modelled at the sub-grid physics.

The idea of high spin, hence higher efficiency, is a good one and indeed there has been some recent work on this regard (Inayoshi & Ichikawa, 2024 <http://arxiv.org/abs/2402.14706>), and we are mentioning this in the revised version, but we do not have the information to model it and it would be out of the scope of our paper.

We also recall our response to point number 3 regarding the nature of the super-Eddington bursts considered by the CAT simulations. The FABLE suite we looked likewise models their bursts by simply adjusting the allowed upper bound (and not just randomly triggering bursts) and are thus not entirely 'ad hoc'. We have added some text to clarify this throughout the text.

6) P9: "the majority of these dormant black holes are expected to be undetected"

Interesting point. Considering the volume sampled by the JADES data, what are the chances to find such a massive BH?

While no real BH mass functions from JWST data yet exist, some efforts were made by Matthee+23 and Kokorev+24, albeit both very uncertain. Also, Kokorev+24 and Dayal+24 discuss the potential for a large number of dormant BHs present at the massive end of the mass function.

I would also look at He+19 mass function albeit it is at lower- z , but could be useful to think about the number densities.

The logarithmic number density of dormant BHs with similar masses to our object derived from the CAT simulations is about -4.9 Mpc^{-3} at $z \sim 6.5$, predicting 1 - 2 such objects across the entire volume sampled by JADES (Trinca+23). This is in line with our approximately derived number density of -5.22 Mpc^{-3} using the detection in GOODS-N. Both of these values are closer to order of magnitude estimates, thus we prefer not to spend too much time discussing them in text aside from a brief mention that has been added in the revised version.

The estimates from Matthee+23 and Kokorev+24 are based on the highly accreting BHs, so they are based on assumptions on the duty cycle.

Regarding Dayal+24, they do not seem to provide a BH mass function and, also, not to provide the fraction of dormant BHs.

7) Figure 5: Zooming in on the Hbeta line I noticed that it does not look symmetric, with a broader wing on its right side. Have the authors entertained why that might happen?

Indeed we have noticed the apparent feature, however it is only marginal: the residual of the (single component) fit in that area is $< 2 \text{ sigma}$. So we prefer not to speculate much about it. However, we agree that it is something to investigate with additional observations. A comment was added in the text to clarify this. Please note that Figure 5 is now Figure 4 in the revised version.

8) I think somewhere in the data section, both PRISM and medium res spectra should be shown alongside one another.

This is now done in Fig. 4.

9) P16: "We therefore assume, as found for other AGN at high- z , that the bulk of the obscuration towards the Broad Line Region also affects the narrow components"

Fair enough. I understand how it can be tough to measure dust extinction in these cases. As much as it is unclear to me if narrow lines can be used instead of broad lines here (and if narrow lines are NLR or galactic), I think the assumptions in this work are clearly stated.

What I propose the authors to carry out, is to see if this A_v estimate holds for the other (narrow) line ratios, like Hb/Hg and Ha/Hg. While Hg is blended with the OIII line checking whether the dust extinction you get from a different ratio of Balmer lines holds, would give more confidence on this estimate.

It is reassuring to see that the Ha/Hb dust extinction is consistent with the one from the optical continuum slope.

Estimations of A_v using the Hg line ratios are uncertain, however, fully consistent with our original estimate giving the values being $A_v = 2.6 +2.6 -2.1$ from Hb/Hg ratio (the high errors likely coming from the proximity of the lines) and $A_v = 2.31 +0.81 -0.61$ from Ha/Hg, assuming case B ratios. A note regarding this has been added to the Methods (sect. 1.4).

Minor:

1) The beautiful spectrum authors show in Figure 5 is one of the main characters of this story. I feel it would be quite nice to show it in the main text instead. I leave this completely up to the authors however.

We thank the referee for the nice suggestion. We have combined the prism spectrum together with the zoom on the Halpha line in Fig 1. The grating spectrum is quite noisy and does not add any extra information besides the Hbeta and [OIII] lines, thus we overlay it in Fig 4 (formerly Figure 5).

2) Page 3: Please write the uncertainty on zspec here.

Uncertainty included.

3) P3: "However, the resolution at this wavelength ($R \sim 460$ for a compact source)". Can the authors quote the rest-frame velocity resolution here too.

Done.

4) Same line: What is the scaling factor assumed here? I believe it is generally assumed to be roughly ~ 1.3 for point-like sources. Please reference the relevant work (or works) where this was first addressed (i.e. de Graaff+23).

The scaling factor ranges from 1.60 to 2.00 across the wavelength range and is close to the lower bound at 5um. We note this factor is not assumed, it is calculated from the optics and light distribution in the slit.

5) P3: "(whose narrow component is stronger than H α)". The narrow part of H α presumably? Please clarify.

Clarified. Thank you for spotting this.

6) P3: "with a systematic scatter of 0.33 dex". I believe "systematic" and "scatter" are two mutually exclusive terms. What I believe the authors mean here is that they take into account

the scatter of both their data and the relation used to obtain M_{bh} . Very nice that you consider and acknowledge this scatter regardless.

This also appears in a few more sentences throughout the manuscript. Double check please.

These instances of 'systematic scatter' have been changed to 'intrinsic scatter'.

7) P3: "reasonably solid" I would change "reasonably solid" to something that inspires a bit more confidence.

Also is it possible to fit the spectrum together with the continuum to try and infer the black hole mass from the underlying 5100 Å continuum? Although if the BH is only 2 % Eddington I would expect the contribution to the continuum to be quite negligible.

Indeed, at this low accretion rate the contribution to the continuum from the host galaxy is significant, about 90% as inferred from the images and from the resulting SED decomposition Fig.13. Therefore, AGN continuum is highly uncertain.

However, we have tried to estimate the BH mass from $L(5100)$ using the methods from Greene and Ho 2005 which comes out to $\log M(BH) = 8.1 \pm 0.8 M_{\text{sun}}$ (the error includes the uncertainty on the AGN continuum level, reddening correction and intrinsic scatter of the virial relation). This is consistent (within the large uncertainty) with the broad Ha estimate. However, given the very large uncertainty, we prefer not to use this estimate in the paper and report it simply in the methods for completeness.

8) P3: ForcePho reference

Added reference to B. Johnson, in prep.

9) P4: "The 8 bands photometry", bands-> band

Corrected, thanks.

10) P4: "These fits, averaged together," Does it mean that they show more or less consistent results?

Both give values consistent within 1 - 2 sigma, as discussed in the methods. A short phrase has been added on P4 clarifying this.

11) P4: "instantaneous star formation rate", how instantaneous? 10 Myr or so I suppose.
Indeed, it is 10 Myr. Clarified in text.

12) P6: "note that our finding of such". "Discovery" would sound better in this context.

Has been changed, thanks.

13) P7: "(as for other high-z AGN)"-> as compared to other high-z AGN?

Has been changed.

14) P7: "massive seeds". a few more words saying explicitly what DCBH entails could be useful. Something about clouds of pristine gas facilitating direct collapse.

Some additional text has been added.

15) P10: "AGN observed at high redshift are seen accreting at super-Eddington". Add Fujimoto+22 ($z \sim 7.2$) to that list.

Added.

16) P10: I am not sure what the authors mean by "tier" here.

A short description of the tiers of JADES spectroscopic survey has been added.

17) Sect 1.1.1. Very first line. "At" -> "in"

Changed.

18) Same paragraph. "The data was processed"-> the data were processed.

Changed.

19) Fig 9: What is the feature at 3.775 μm ?

Likely an artifact that survived our sigma clipping. This has been clarified in the caption. In the revised version Figure 9 is Figure 8.

20) P17: "galaxy component appears bright enough for reliable photometry."
Please be more specific here. Quote average S/N per pixel/resolution element? Or just the average S/N in your filters.

The fitting procedure used here does not utilize the final reduced images. Instead it computes a model to fit individual exposures, thus estimating the S/N for the final galaxy component is not entirely straightforward. A good estimate can be performed by dividing the model pixel fluxes by the background rms in each pixel, giving S/N values ranging from 6 to 40 across our filter range, which we quote there in text.

21) P17: "illustrates" -> illustrate

Changed.

22) P17: "galaxy contains significantly more flux". I would rephrase this slightly to read a bit better. Something like "galaxy component is dominated by blue light...". Saying this in this particular way also may start thinking about the blue component found in LRDs and its uncertain (as of yet) origin.

Has been rephrased.

23) Fig 11: Can you add another row showing the AGN model by itself?

Done.

24) Fig 11: Consider showing the observed source magnitudes and the uncertainty for the top panels here.

Done.

25) Fig 13: The dynamic range of the plot is a bit too wide to see how well the points fit here. As the lines are irrelevant here I suggest to the authors to narrow it down a bit, it's fine to cut off the lines.

Done.

26) Fig 13: Can you show the reduced chi2 here in addition or instead of the total one?

Done.

Referee #2

- Figure 11 shows the JWST image of this object in multiple filters, and illustrates the process of separating the central point source and the extended galaxy. Doing this well (especially given the very small scale size inferred for the host galaxy; see comments below) is quite tricky, and we are concerned that the systematic errors in the process haven't been fully incorporated into the analysis. In particular, small errors in the assumed point spread function (PSF) have the potential to significantly affect the inferred host galaxy properties. The paper needs to describe how the PSF was actually determined, and the robustness of the results to reasonable variations in the PSF in each of the filters. See for example the discussion in Ding et al (2023; reference 34

in the present paper). It is stated at the top of Page 19 that the stellar mass is altered by less than 1 sigma if *no* PSF subtraction is done. Is this equivalent to saying that that the quasar component is close to negligible in the imaging?

The PSF used in ForcePho is approximated as a Gaussian mixture model (GMM). This indeed introduces systematics, which are likely unaccounted for in the purely statistical errors. To test these systematics, we refit the data with a newer GMM approximation of the PSF, which includes accounting for charge transfer effects, and find that the inferred radius becomes 0.021" instead of the previous 0.025" value. We thus adopt the 0.004" as the 1 sigma additional systematic error from the PSF.

However, as pointed out by the referee, indeed, even without PSF subtraction the inferred properties of the host galaxy do not change significantly (specifically, stellar mass and SFR differ by no more than 2 sigma). This is indeed consistent with our finding of the AGN component being sub-dominant (only about 10%) in all bands (except for F444W which includes the dominant contribution from the broad Ha). Therefore, our results are not drastically affected by the detailed knowledge of PSF.

Lastly, the continuum emission of the QSO is likely sub-dominant in the photometry as fitting the fluxes extracted from the combined images without decomposition yields stellar mass and SFR differences of no more than 2 sigma given the uncertainties.

In addition, we perform an additional check of whether a source is indeed extended, and not just a PSF, by measuring the radial profiles from the stacked images in the bluer bands, where the PSF is smaller and the stellar component is expected to dominate even more, based on the initial decompositions. Specifically, the image below shows the object in F115W and F277W bands (left-most column), the PSF in the corresponding bands (middle column) and the derived brightness profiles (right column). The profiles in the F277W band were measured by isophote fitting. As can be seen in the top row, the best fit to our object consists of slightly ellipsoidal isophotes, while the PSF's are circular. We measure the radial profile by placing circular apertures on the centroid. In both cases, as shown in the right-most column, the source, while compact, is significantly larger than the PSF, presenting a compelling reason for the existence of an extended component. We also present this figure (as

Figure 10) and accompanying discussion in the methods.

We finally point out that our target is offset from the local relation by nearly three orders of magnitude. There even if the stellar mass was underestimated by a factor of a few, this would not change our conclusions. In order to be consistent with the local relation, a stellar mass larger by three orders of magnitude would be required, which would be totally inconsistent with the observed SED, even if all emission was ascribed to the host galaxy.

Also, the aspect of the galaxy being quiescent, or below the star forming main sequence (which depends on the results of the SED fitting), is not crucial to the paper, and indeed it has been moved to the Methods. However, it can be entirely removed, if required by the referees.

Lastly, please note that Fig 11 has changed to Fig 12 in the revised version.

- Related to the previous point, the host galaxy stellar mass, size, and star formation rate (which lead to many of the key results of the paper: the large ratio of black hole to stellar mass and the quiescence of the host) all rest heavily on the correct decomposition of quasar and host galaxy light. A more thorough discussion of how you attribute continuum flux to the quasar or galaxy, and the systematic uncertainties in that process, would be appropriate. That is, we are somewhat skeptical of the robustness of the SED fitting given the potential for significant systematic errors on the host photometry. On a related note, the SED of the host and quasar drawn from the imaging (Figures 12 and 13) should be consistent with the shape of the

spectrum itself (Figure 5); is this indeed the case? The spectrum has a blue continuum (at least in f_λ), while the SED is quite red (in f_ν); are they consistent? Showing that the spectra and SEDs match would go a long way in confirming the robustness of the photometry, although it is not clear if it will give further insights into the quasar/host decomposition.

As mentioned in the previous response, the exact decomposition does not affect the inferred properties significantly as fitting the combined photometry gives similar values. In the figure below, we have overplotted the Prospector SED for the host galaxy on top of the full 1D spectrum from the prism and they appear largely consistent within the error bars (shaded regions).

This also confirms that the spectrum is dominated by the stellar continuum, as found in our decomposition, except for the reddest photometric filter (F444W) in which the AGN flux is boosted by the contribution of the broad Ha. Additionally, we note that lines emission strengths predicted by Prospector differ significantly with the [OII] doublet being weaker and [OIII] stronger than predicted. This is consistent with our interpretation that the continuum is dominated by stellar emission while lines are produced primarily by the AGN.

We are not sure this plot is very useful and, given that we already are showing a lot of figures (as pointed out by the referees) and that we are exceeding the word limit even for the Methods, we would be inclined to leave it out. However, we can find a way to include it, if the referees feel strongly about it.

- Table 2 lists the fit parameters from the imaging and spectroscopy. The effective radius of the host galaxy is given as 137 ± 8 parsecs. At this redshift, this corresponds to $0.025''$, while the diffraction limit at 356W (where Figure 5 implies the host galaxy to be well-defined) is about $0.1''$. It is of course possible to measure scale sizes significantly smaller than the PSF with sufficiently high S/N data and a sufficiently accurate model of the PSF, but we would like to be convinced that this measurement is robust. Note that this is true even if the quasar contribution to the image is negligible, as the galaxy image itself is also broadened by the PSF. We are similarly suspicious of the tiny error bar on the scale size (and the similarly small error bar on the Sersic index n).

Indeed the small uncertainty on the effective radius comes from the statistics of the fit itself. Yet, using the revised systematic uncertainties, following what was discussed in response to the first comment, the systematic error on the radius turns out to actually be 16%. This reevaluates the effective radius estimate and its uncertainty to 137 ± 23 pc. We are grateful to the referees for their comments that prompted us to properly reevaluate the uncertainties.

-Similarly, the dynamical mass of the galaxy depends on R_e , as well as σ_* , measured from the width of the [OIII] line (Figure 9). The line appears to have a full width at half maximum of about 3 pixels, i.e., it is very close to being unresolved. However, there is no discussion of the uncertainty in the value of σ_* due to uncertainties in the resolution of the spectrum, or the fact that the inferred velocity dispersion is in effect the difference in quadrature of the measured line width and the resolution of the spectrograph. The quoted uncertainty on σ_* is 6 km/s, which seems unrealistically small. This leads to an extremely small uncertainty on the dynamical mass of the galaxy, 0.04 dex; we are not at all convinced of this.

The systematic uncertainty on the broadening by the LSF is of order 10 - 20% (see de Graaf et al. 2023). When included in the analysis, the error on the σ_* is increased to ~ 16 km/s. The uncertainty on M_{dyn} increases similarly, to ~ 0.2 dex. However, the calibration uncertainties on the relations used for the stellar velocity dispersion and dynamical mass have an intrinsic scatter of 0.3 dex, which ends up being dominant in the error budget. However, the discussion on the dynamical mass is secondary in the paper and can be entirely removed if the referees feel strongly about it.

-The black hole mass estimate is based on a calibration from low redshift; it should be stated explicitly that one has to assume that this calibration holds at this high redshift. In addition, the low-redshift calibration itself is known to have an intrinsic scatter of order 0.3 dex, which should be included in the error budget.

The scatter has been included in the error budget. We have also added a short statement clarifying our assumption for the local virial relation being valid at high- z . The paper already

discusses the validity of local relations, at least when using the same diagnostic (H α), in the context of the recent GRAVITY results.

These are our major concerns. The following are somewhat smaller issues in the paper, that we encourage the authors to consider. Reading sequentially through the paper:

-Page 6: You state that the existence of this source is "incompatible" with the selection effect scenarios. This seems like a strange way to phrase it; perhaps the statement is that the data are deep enough to be less sensitive to selection effects (as you show in Figures 4 and 18).

Clarification added.

-The upper left panel of Figure 5 seems to show that [OIII] and H α are extended. Is this indeed the case? If so, one can get a measure of star formation rate in the host galaxy which is independent of that determined from the SED fitting.

Indeed, both [OIII] and H α lines appear to be extended. However, due to the compact size of the object, which is comparable to NLR of Seyfert 1 galaxies, the extended portion of those lines is likely still contaminated by AGN photoionization. Therefore, star formation rates inferred from them would likely bias high. Indeed, we have attempted to estimate the SFR using the flux of the full narrow H α and came away with a value of ~ 15 Msun/yr; however, the narrow lines of our object are likely produced by the AGN as their ratios match AGN photoionization models as shown by Mazzolari et al 2024 (<https://arxiv.org/pdf/2404.10811>). Therefore the star formation rates obtained from them are likely conservative upper limits and not representative of the real value. Please note that Figure 5 is Figure 4 in the revised version. Once again, the SFR of the host galaxy is not crucial for the main conclusions of the paper, and indeed has been moved to the Methods. However, we can even totally remove it, if the referees require it.

-Figure 6 shows that the object was not properly centered in the slit. If the host galaxy is rotating, the width of the [OIII] line used for the dynamical mass estimate will be underestimated, as part of the rotation curve is systematically cut off. This is worth some discussion.

This is a good point. We have included a few sentences on this in the section on the BH-sigma relation and dynamical mass measurement. Please note that Figure 6 is Figure 5 in the revised version.

-This paper has a total of 18 figures, some of which seem less necessary to show. We suggest dropping Figure 7; Figure 18 is quite similar to Figure 4, and perhaps is not needed either.

We appreciate the referee's suggestion, however Figure 7 (Figure 6 in the current version) showcases the posteriors from our fitting procedure and provides an accessible visual proof that our fit has converged and does not contain significant degeneracies or secondary solutions. Therefore, we would prefer to keep it in the methods section for this purpose. As for Figure 18 (Figure 19 in the current version), it is indeed similar to Figure 4 (Figure 3 in the revised version), however, it displays the results of an independent suite of simulations and it is therefore useful to illustrate what is discussed in the text in that section. Of course, we are open to their removal if the referee or the editor feel strongly about it.

-Page 13: You explore the possibility that the object includes a supernova. You should add a brief description of why this possibility is being considered; is the idea that the broad H alpha could be due to a supernova in a nebular phase?

Indeed. We have added a short comment to clarify it.

-Page 16: The best-fit extinction is $A_V = 2.0 \pm 0.4$ magnitudes. But you say that the bolometric luminosity changes by less than 2 sigma if you ignore extinction. That's puzzling, as the extinction seems to be detected at 5 sigma (and bolometric luminosity should scale directly with the extinction). This requires a bit more explanation.

Indeed, the bolometric luminosity changes by ~ 1 dex and is significant, however, the changes on the BH mass and Eddington ratio are only ~ 0.5 dex each, which correspond to a roughly 2 sigma change given the uncertainties. We have edited the text there to clarify that we mean M_{BH} and λ_{Edd} specifically. Thank you for spotting the improper phrasing.

-Figure 9 shows a piece of one of the three medium-resolution grating observations of this source. Did the other two include any emission lines?

Note that the y-axis units in Figures 9 and 5 are the same, but labeled differently (10^{-16} erg/s/cm²/micron vs. 10^{-20} erg/s/cm²/Å). However, the strength of the spectral features in the two spectra do not seem to be consistent with each other. This should be made consistent.

Currently, in the revised version, we are showing the less noisy part of the R1000 spectrum together with the prism in Figure 4 (former Figure 5). As it can be seen there, the R1000 does not contain any emission lines detected with high S/N aside from [OIII] and Hbeta. The y-axes of all spectra are now in consistent units of 10^{-16} erg/s/cm²/micron. It should be noted that, obviously, spectrally unresolved lines do appear sharper and with higher peak intensity in higher resolution gratings spectra, as the spectral pixels size is smaller. In our case, the narrow lines are only marginally resolved even in R1000, thus they appear higher there than in prism, however, the measured fluxes are fully consistent across the two dispersers. Additionally, please note that Figure 9 is Figure 8 in the revised version.

-The caption to Figure 11 should state what the spatial scale of each panel is.

Comment added. Note that Figure 11 is now Figure 12.

-It would be appropriate to include the F356W and F410M bands in Figure 13 (perhaps with a different symbol) even if they are not used in the fit. It would be interesting to see if the effect of the emission lines is directly seen in the photometry.

The filters are now included in the figure (Figure 14 in the revised version) using different colored symbols and show clear excess above the remaining bands. We thank the referee for this nice suggestion.

Page 21, Section 2: given the large inferred gas mass, it would be interesting to speculate on whether molecular gas might be detectable in this source with ALMA.

While such speculation could certainly be useful in planning an ALMA followup, our source is unfortunately located in the GOODS-N field and is thus inaccessible to ALMA observations. NOEMA does not have the sensitivity to detect such gas masses; indeed, the estimated gas mass appears large but much lower than typically observed at $z \sim 2$ in unlensed fields in NOEMA deep programmes (e.g. the PHIBSS survey by Tacconi and collaborators, which are sensitive to molecular gas masses $> 10^{10} M_{\text{sun}}$).

Table 2: We have already commented on the surprisingly small error bars of many of the tabulated quantities. We also point out that the value of T_e is written with considerably too many significant figures; perhaps it should be, $25400 \pm 3200 - 4800$ K.

As commented above, the errors have been re-estimated. Additionally, the significant figures have been adjusted.

We would like to thank the referees for providing helpful comments that aided in the process of getting the paper ready for the final submission.

Referee 1

A few small things.

1) Your last bold text on Page 5 says:

"These predict that either black holes are born from relatively massive seeds (e.g. direct collapse black hole, originating from clouds of pristine gas) accreting below the Eddington rate, or from either light seeds (stellar remnants) or heavy experiencing short phases of super-Eddington accretion ([11–17, 32–35]."

I think the "or heavy" part here is redundant since the DCBH are already the heavy seeds.

Our intention here was to highlight the different scenarios in CAT models. In the Eddington-limited case, only heavy seeds reach the masses required, while in the super-Eddington case both seeding models can reach the required masses. We have edited the sentence to hopefully highlight this more clearly.

2) I think it would be better to replace R1000 in Table 1 to G395M. Same in the caption of Fig.4 and the legend of the figure. Up to you though.

We appreciate the suggestion, however, we feel that labeling the data according to its spectral resolution is somewhat more informative of the physics than labeling it by instrumental configuration. Yet if the referee and/or the editor feel strongly about it then we are certainly open to change it.

3) Section 1.1.1. When you list dispersers and filters as "(G140M/F070LP, G235M/F170LP, G395/F290LP)", the G395M is missing an M.

Thank you for spotting it, fixed.

Referee 2

Referee #2 (Remarks to the Author):

We appreciate the effort the authors have put into addressing our and the other reviewer's concerns, and we find the revised manuscript to be an interesting and thorough work with relevance to the current

field. We think the paper should be accepted for publication with only minor updates and an additional editing pass. We do have some remaining concerns, but they are minor enough that there is no need for us to see the paper again before it is accepted.

Reading sequentially through the paper:

Page 6: The paper could use another editing pass to tighten up the discussion and remove repetition. This was particularly apparent on Page 6, where the addition of several sentences to respond to the referees' concerns makes the discussion choppy and repetitive. For example, super-Eddington accretion is mentioned in three paragraphs in a row, each time making essentially the same point. And the very long paragraph starting, "Finally, we argue that dormant..." also needs tightening.

We have revised the paragraphs mentioned, trimming and rearranging the text to improve the flow. We thank the referee for these suggestions.

Figure 4 (page 9): The figure caption should explain the relationship between this figure and the similar Figure 1.

Thank you for the suggestion. A short comment added.

Page 10 is another example of a repetitive and overly lengthy discussion about the presence of a broad component in H beta.

We have tightened the discussion there to be more succinct.

Page 14 describes the PSF modeling and the uncertainties thereof. In the response to our report, the authors stated that the PSF is modeled with a Gaussian Mixture Model, but it is not stated either in the response or the revised paper to which data this model was fit. Stars in the field of view? An a priori model based on the optics of JWST? This should be clarified. In addition, it is stated that a different PSF model which incorporates charge transfer effects allows one to understand the systematic error in the size of the host galaxy. Without knowing more about the PSF modeling process, the reader cannot assess whether this is a full accounting of the PSF uncertainty.

The PSF model used by ForcePho is based off of the webbpsf models, which are a priori based on models of the instrument's optics and electronics. The final form of webbpsf models used are also calibrated to better match observational results from field stars. We have added a short sentence clarifying the above.

We appreciate the addition of Figure 10 and its demonstration of the extent of the galaxy!

Page 16: It is stated that the F277W band "smears the source over more pixels". Is the point simply that the PSF is largest in this band? In the next paragraph, in F115W "the flux was dispersed over too few pixels". Is this a statement that the PSF is somewhat undersampled?

The PSF is the largest in the F444W band, but that one contains the H α emission and is expected to be dominated by AGN emission. The F277W PSF is larger than F115W and is thus better sampled at the same pixel scale, while F115W is undersampled. We amended the text to clarify this.

We have a general concern that the paper may be somewhat too long.

This will be a question for the editor to address, but tightening the text in various places (some of which are indicated above) will help. Other areas which felt overly long were:

-The discussion of the extinction on the host galaxy. Multiple methods are used to estimate the extinction. All have very large error bars, and it is not surprising that within these uncertainties, they agree with one another. Quite a bit of text is devoted to saying this; it could be stated more succinctly.

-Section 5 on the future evolution of this system: this seems like pure speculation that doesn't directly support anything in the main paper, and thus seems out of place in the supplementary materials section of the paper.

We have amended the section on the A_V estimates to be shorter and clearer. We have followed the referee's suggestion by removing Section 5.